

# A tropopause-based a priori for IASI-SOFRID Ozone retrievals: improvements and validation

Brice Barret[1], Emanuele Emili[2], and Eric Le Flochmoen[1]

[1]Laboratoire d'Aérologie/OMP, Université de Toulouse, Toulouse, France.
[2]CECI, Université de Toulouse, CERFACS, CNRS, Toulouse, France.

**Correspondence:** B. Barret
(brice.barret@aero.obs-mip.fr)

**Abstract.**

The Metop/IASI instruments provide data for operational meteorology and document atmospheric composition since 2007. IASI Ozone ($O_3$) data have been used extensively to characterize the seasonal and interannual variabilities and the evolution of tropospheric $O_3$ at the global scale. The SOFRID (SOftware for a Fast Retrieval of IASI Data) is a fast retrieval algorithm that provides IASI $O_3$ profiles for the whole IASI period. Up to now SOFRID $O_3$ retrievals (v1.5 and 1.6) were performed with a single a priori profile which resulted in important biases and probably a too low variability. For the first time we have implemented a dynamical a priori profile for spaceborne $O_3$ retrievals which takes the pixel location, time and tropopause height into account for SOFRID-O3 v3.5 retrievals. In the present study we validate SOFRID-O3 v1.6 and v3.5 with ECC ozonesonde profiles from the global WOUDC database for the 2008-2017 period. Our validation is based on a thorough statistical analysis using Taylor diagrams. Furthermore we compare our retrievals with ozonesonde profiles both smoothed by the IASI averaging kernels and raw. This methodology is essential to evaluate the inherent usefulness of the retrievals to assess $O_3$ variability and trends. The use of a dynamical a priori largely improves the retrievals concerning two main aspects: (i) it corrects high biases for low-tropospheric $O_3$ regions such as the southern hemisphere (ii) it increases the retrieved $O_3$ variability leading to a better agreement with ozonesonde data. Concerning UTLS and stratospheric $O_3$ the improvements are less important and the biases are very similar for both versions. The SOFRID Tropospheric Ozone Columns (TOC) display no significant drifts (< 2.5%) for the northern hemisphere and significant negative ones (9.5% for v1.6 and 4.3% for v3.5) for the southern hemisphere . We have compared our validation results to those of the FORLI retrieval software from the litterature for smoothed ozonesonde data only. This comparison highlights three main differences: (i) FORLI retrievals contain more theoretical information about tropospheric $O_3$ than SOFRID (ii) RMSDs are smaller and correlation coefficients are higher for SOFRID than for FORLI (iii) in the northern hemisphere, no significant temporal drift is detected in SOFRID contrarily to FORLI ($\sim$8%).



## 1 Introduction

Ozone ($O_3$) in the stratosphere protects life from solar UV radiation. Close to the surface, $O_3$ is an oxidative pollutant harmfull for human health through irritation of respiratory tract (Brunekreef and Holgate, 2002) and for vegetation through deposition on leafs that leads to the reduction of plant growth (Ainsworth et al., 2012). Tropospheric $O_3$ is also a powerful greenhouse gaz

which increase during the 20-th century has significantly contributed to global warming (Shindell et al., 2006). The radiative forcing of $O_3$ is particularly important in the tropical Upper Troposphere-Lower Stratosphere (UTLS) (Chen et al., 2007).

It is therefore important to document the evolution of $O_3$ in these different layers independently. There are clear evidences from satellite databases that upper stratospheric $O_3$ has increased since 1997 following the ban of CFC's by the Montreal protocol (Ball et al., 2018). Nevertheless, the total column $O_3$ is stable since 1998. According to Ball et al. (2018) this con-

tradiction is due to the fact that lower stratospheric $O_3$ is declining and compensates both stratospheric and tropospheric $O_3$ increase. Based on OMI/MLS Tropospheric Ozone Columns (TOC) they state that TOC is globally increasing. OMI/MLS data for the 2005-2016 period are indeed documenting global positive TOC trends with particularly large increases over Asia (Ziemke et al., 2019). Based on 10 years of retrievals with the Fast Optimal Retrievals on Layers for IASI $O_3$ (FORLI-O3) software, Wespes et al. (2018) document a decrease in tropospheric $O_3$ levels in the Northern Hemisphere (NH). Another

IASI tropospheric $O_3$ product (KOPRAFIT-O3) displays a TOC decrease over continental China (Dufour et al., 2018). In their exhaustive work about TOC evolution, Gaudel et al. (2018) clearly highlight the contradiction between global increase (OMI/MLS and other UV-Vis products) on the one hand and global decrease (IASI) on the other hand. They also show that the different satellite products agree on a TOC increase over Asia. Among the two global IASI TOC datasets used in Gaudel et al. (2018), FORLI-O3 is indicating a significant global decrease and $O_3$ retrievals with the Software for a Fast Retrievals of

IASI Data (SOFRID) a slightly weaker and less significant one. Two versions of FORLI-O3 have been validated by Boynard et al. (2016) (v20141022 ) and Boynard et al. (2018) (v20151001). They both document a problem (drift or jump) in the $O_3$ retrievals around year 2011 but this does not hinder the fact that TOC are decreasing according to Wespes et al. (2017). It has to be noted that both validation studies compare IASI retrievals to ozonesonde profiles smoothed by the retrieval averaging kernels. Such a comparison enables the detection of abnormal biases, variability or drifts in the retrievals but do not document

the ability of FORLI-O3 to reproduce real $O_3$ levels and variabilities. SOFRID-O3 has only been validated at the beginning of the IASI period on a very short time period (Barret et al., 2011) and on a longer time period together with FORLI-O3 and KOPRAFIT-O3 (Dufour et al., 2012). Furthermore, EUMETSAT L2 atmospheric temperature poducts retrieved from IASI and used for FORLI (v20141022 and v20151001) and for SOFRID-O3 v1.5 retrievals are not stable in time (Boynard et al., 2018). Therefore we have reprocessed the whole IASI database using ECMWF operational analyses for temperature and humidity to

produce SOFRID-O3 v1.6. SOFRID-O3 has been shown to overestimate low troposphric ozone over the southern hemisphere (SH) (Dufour et al., 2012; Emili et al., 2014, 2019). Emili et al. (2014) have hypothetized that this overestimation was due to the use of a single a priori profile biased towards northern hemisphere (NH) mid-latitudes $O_3$. In order to verify this hypothesis and to improve our $O_3$ retrievals, we have developed a new version of SOFRID-O3 (v3.5), with a dynamical a priori profile



based on a global $O_3$ climatology (Sofieva et al., 2014).

The aim of the present paper is to validate both SOFRID-O3 latest products (v1.6 and v3.5) on the whole IASI period (2008-2017) in order to infer their ability to reproduce tropospheric $O_3$ levels and variability on seasonal to decadal time
scales. The validation is based on $O_3$ profiles from ozonesonde retrieved from the World Ozone and Ultraviolet Radiation Data Centre (WOUDC) database. In section 2 we describe the characteristics and differences of SOFRID-O3 v1.6 and v3.5 retrievals. Section 3 is dedicated to the description of the validation methodolgy based on comparisons between smoothed and raw ozonesonde data and we provide our validation results in section 4. Based on (Boynard et al., 2018) we also compare our results to FORLI-O3 (section 5) before concluding in section 6.

## 2   IASI SOFRID-O3 Retrievals

IASI is a spaceborne thermal infrared nadir spectrometer. IASI has a moderate spectral resolution combined with a high signal to noise ratio and a 12 km footprint at nadir (Clerbaux et al., 2009). Thanks to its large accross track scanning ($\sim$ 2200 km), IASI revisits each scene twice daily around 9:30 solar time in the morning and in the evening. Three IASI instruments have
been launched on the Metop meteorological platforms (Metop-A in 2008, Metop-B in 2012 and Metop-C in 2018). Here we present results based on $O_3$ retrievals from 10 years of Metop-A/IASI data. We will present results based on the morning over-pass data only as they are known to provide more information that nighttime data. Furthermore, it facilitates the comparison to other validation studies (Boynard et al., 2016, 2018) also based on morning data.

The SOFRID software first described in Barret et al. (2011) is based on the RTTOV (Radiative Transfer for TOVS) operational radiative transfert code (Saunders et al., 1999; Matricardi et al., 2004) combined with the 1D-Var software (Pavelin et al., 2008) both developed within the framework of EUMETSAT NWP-SAF. The $O_3$ profiles are retrieved from the 980-1100 cm$^{-1}$ spectral window encompassing the 9.6 $\mu$m $O_3$ absorption band. Only cloud free or weakly contaminated pixels are processed with clouds filtered using AVHRR-derived fractional cloud cover or a test based on brightness temperatures at $11\mu$m and $12\mu$m
when AVHRR cloud cover is not available as described in Barret et al. (2011). The two SOFRID-O3 versions that are validated and compared in the present paper have significant differences that are described below.

### 2.1   Single a priori profile: V1.6

SOFRID-O3 V1.6 is almost similar to V1.5 described in Barret et al. (2011). It is based on RTTOV V9.3 (Saunders et al.,
1999). In RTTOV, the optical depths are expressed as a linear combination of profile dependent predictors that are functions of temperature, absorber amount, pressure and viewing angle. In RTTOV V9.3, the regression coefficients are derived from computations with the LBL Radiative Transfer Model V11.6 (LBLRTM Clough et al. (2005)) on 43 atmospheric levels using



the HITRAN2004 spectrocopic database (Rothman and Jacquemart, 2005). The single difference is that V1.6 uses temperature and humidity profiles from ECMWF operational analyses for the RTTOV simulations and V1.5 was using IASI L2 products delivered by EUMETSAT. The change has been operated for availability problems and mostly because the EUMETSAT L2 products are not homogeneous over the whole 2008-2017 period which could result in retrieval inconsistencies (Boynard et al., 2018). We use 6 hourly ECMWF analyses which are provided on 91 (resp. 137) vertical levels until (resp. after) 24 June 2013 from the ground up to 0.02 hPa on a 0.25°x0.25° horizontal grid. The ECMWF temperature and humidity profiles are interpolated to the time and location of the target IASI pixel with a 3D-linear interpolation scheme.

$O_3$ concentrations are retrieved on the 43 RTTOV levels with the NWPSAF 1D-Var algorithm (Pavelin et al., 2008) based on the Optimal Estimation Method (Rodgers, 2000). The OEM is a Bayesian method where the incomplete information provided by the measurement is complemented by a priori information which is supposed to represent the best knowledge of the state vector at the moment of the measurement. In our case the state vector is the $O_3$ profile. For both V1.5 and V1.6 we use a single $O_3$ a priori profile which is based on two years (2008-2009) of WOUDC and MOZAIC-IAGOS profiles completed to the top of the RTTOV V9.3 model (0.1 hPa) by MLS averaged profiles (see Barret et al. (2011) for details).

## 2.2 Dynamical a priori profile: V3.5

As V1.6, SOFRID-O3 V3.5 uses interpolated temperature and humidity profiles from ECMWF analyses. It is based on the more recent RTTOV V11.1 (Hocking et al., 2015) which regression coefficients are derived from LBLRTM V12.2 computations on 101 vertical levels with the HITRAN2008 spectroscopic database (Rothman et al., 2009). The second and more important one is that it uses dynamical a priori profiles from TpO$_3$, the $O_3$ profile tropopause based climatology of Sofieva et al. (2014). This climatology is based on ozone profiles resulting from merging ozonesonde data in the troposphere and SAGE-II V6.2 data (Wang et al., 2006) in the stratosphere. The ozonesonde profiles (36000) extracted from the Binary Data Base of Profiles (BDBP) come from 136 stations for the period 1980 to 2006 (Hassler et al., 2008). For each merged ozonesonde-SAGE-II profile, the tropopause was computed according to the the World Meteorological Organization (WMO) definition of the lapse-rate tropopause (WMO, 1957). For each month, the ozone profiles are gathered according to 10° latitude bins, 1 km tropopause intervals and the corresponding averaged profiles together with their $1\sigma$ variabilities are computed and provided. Variable a priori profiles have already been used for satellite sensor retrievals. For instance, TES $O_3$ retrievals used monthly mean profiles from the MOZART CTM averaged over a 10°latitude x60° longitude grid (Bowman et al., 2009). OMI $O_3$ a priori profiles are based on monthly and latitude dependent ozone profile climatology (McPeters et al., 2007) derived from ozonesonde and satellite data (Liu et al., 2010). Nevertheless, the use of an a priori simply based on the geographical location of the satellite pixel does not allow taking the atmospheric dynamics into account. For instance, at a mid-latitude location, the $O_3$ profile can be typical of mid-latitudes one day and polar (low tropopause) or tropical (high tropopause) a few days later depending on the global atmospheric dynamics (position of the polar or subtropical jets, anticyclones). The use a Tropopause dependent climatology allows us to take the atmopsheric dynamics into account and provides a much more accurate a priori $O_3$ profile.





This technique was once used for $O_3$ total column retrievals from FTIR spectra at the Jungfraujoch station (De Maziere et al., 1999). It was shown that the retrieved $O_3$ columns were largely improved when the tropopause was taken into account in the choice of the a priori. In SOFRID-O3 V3.5, we compute the tropopause using the WMO lapse rate definition from the ECMWF interpolated temperature profiles. The a priori profile is then picked up from the $TpO_3$ climatology according to month, latitude and tropopause height.

## 2.3 Information content and retrieval error

A remote sensing instrument is not equally sensitive to the different atmopsheric layers. Its vertical sensitivity depends on its instrumental characteristics and on local parameters. In the case of a thermal infrared nadir sounder such as IASI, surface parameters such as surface emmissivity, surface temperature, thermal contrast between the surface and the first atmospheric layer are key parameters to determine the vertical sensitivity, especially in the lower troposphere (Barret et al., 2005; Boynard et al., 2016). The vertical sensitivity of a remote sensing instrument is characterised by the so-called Averaging Kernel (AK) matrix. For each retrieval layer, the retrieved quantity is the result of the convolution of the whole real profile by the corresponding averaging kernel (row of the AK matrix) plus a contribution from the a priori profile ($\mathbf{x}_a$) and a noise ($\epsilon$) contribution (see Eq. 1).

$$\hat{\mathbf{x}} = \mathbf{A}\mathbf{x} + (\mathbf{I} - \mathbf{A})\mathbf{x}_a + \mathbf{G}(\epsilon) \tag{1}$$

In an ideal case, the AK matrix ($\mathbf{A}$) would be the identity matrix ($\mathbf{I}$) and real ($\mathbf{x}$) and retrieved ($\hat{\mathbf{x}}$) profiles would be identical modulo the noise ($\epsilon$) contribution. In a real case, the AKs are bell shaped functions which peak at an altitude that could be different from the nominal altitude and which width gives an indication of the retrieval vertical resolution.

The Degree of Freedom for Signal (DFS) of a retrieval describing the number of independent pieces of information provided by the measurement is the trace of the AK matrix (Rodgers, 2000). We have divided the atmosphere in 5 layers which are described in Table 1. The Troposphere 2 layer has been selected for comparison with Boynard et al. (2018, 2016) who did not compute a tropopause based TOC for their validation (see section 5). The DFS corresponding to these different layers is displayed in Figure 1 for V1.6 and V3.5 averaged over the validation dataset. The total DFS ranges from 2.4 to 3.3 for v3.5 and is about 0.2 lower for v1.6. The DFS for the troposphere (WMO lapse rate), UTLS and stratosphere are almost identicals for both versions. The tropospheric DFS is the lowest (0.3-0.5) at high latitudes where surface temperature, thermal contrast and tropopause height are the lowest and the highest in the tropics (about 1.5) where surface temperature and tropopause height are the highest. At mid-latitudes the tropospheric DFS is about 0.6. Therefore, except in the tropics, SOFRID retrievals provide less than one independent piece of information in the troposphere. In the UTLS (resp. stratosphere) the DFS range from 0.7 to 1 (resp. from 0.9 to 1.5) which means that SOFRID provides around one independent piece of information in these layers.





Th retrieval error is the sum of the measurement and smoothing errors (Rodgers, 2000). Uncertainties in auxiliary parameters (Temperature and humidity profiles, surface properties...) are also responsible for errors. Coheur et al. (2005); Barret et al. (2005) have shown that in the case of $O_3$ and CO retrievals from thermal infrared satellite sensors the dominant source of errors was the smoothing error. The retrieval error for SOFRID-O3 v1.6 and v3.5 are displayed in Fig. 1. V1.6 displays slightly larger

errors than v3.5 but the same behaviour. For the Total and stratospheric columns, the errors decrease from high latitudes (9-12 DU) to the tropics (6-8 DU). The behaviour of UTLS errors is similar with lower values (4 to 6 DU). For the TOC, errors are larger in the tropics (5 DU) than at middle and high latitudes (4 DU).

## 2.4 Global distributions of tropospheric ozone columns

The global distributions of TOC from SOFRID v1.6 and v3.5 for July and December 2017 are displayed on Fig. 2. The global TOC structures are similar for both versions. They both clearly show the highest TOC over the NH mid-latitudes in summer with a large export region over the north Pacific off the chinese coast and the summertime TOC maximum over the Eastern Mediterranean already documented with the GOME-2 sensor (Richards et al., 2013). The tropical Wave-one pattern (Thompson et al., 2003; Sauvage et al., 2006) with the highest TOC over the tropical Atlantic and the lowest one over the South

Pacific Convergence Zone (SPCZ) is also noticeable for both versions. Sauvage et al. (2006) have shown that the tropical Atlantic maximum was mostly a result of African and South American Lighning NOx (LiNOx) emissions. High TOC are also detected during austral summer over southern Africa and the southern Indian Ocean towards Australia. According to Zhang et al. (2012), these high TOC are mostly caused by LiNOx emissions from central Africa with a yearly maximum in May. The clearest difference between both versions is that v3.5 produces lower TOC than v1.6 in the low tropospheric $O_3$ regions. This

is clear over the Inter Tropical Convergence Zone (ITCZ) and the SPCZ, over the SH for both seasons and over the NH mid latitudes in winter. We will show in the validation part of the paper that this is an important improvement of the SOFRID-O3 retrievals. The agreement is better in regions of high TOC such as NH mid latitudes in summer or the tropical Atlantic.

The use of a dynamical a priori is responsible for visible stripes along the 10 latitude bands. These stripes are generally

indicating a discontinuity of 2.5 to 5 DU between two adjacent latitude bands with different a priori profiles. They are clearly caused by the impact of the a priori on the retrieval which is taken into account in the retrieval error (see Equ. 1). The latitudinal discontinuities are therefore consistent with our retrieval errors (4-5 DU) from Fig. 1.





# 3 Validation methodology

## 3.1 Ozonesonde data

Ozonesonde data come from the WOUDC database (hhttps://www.woudc.org/). For consistency purposes we have chosen to use data from ECC sondes only. For the 10 years IASI period (2008-2017), valid comparisons were effective for about 12000

ozonesonde profiles among the 16000 downloaded. A map with the number of sondes used for the validation at each station over the 2008-2017 period is displayed in Fig. 3. Most (∼7000) of the validation sondes were launched in the NH mid-latitudes with 15 stations providing more than 1 profile per month on average (more than 120 profiles for 10 years) mostly in Western Europe and Northern America. For all other 30° latitude bands, the number of validation profiles range from 800 to 1200 with only 3 to 4 stations providing more than 120 profiles. The balloons that carry the ozonesondes explode below 40 km. In order

to complete the ozonesonde profiles in the upper stratosphere and mesosphere, we have used MLS data averaged on 10 days on a 10°x10° grid (see Barret et al. (2011) for details).

## 3.2 Coincidence criteria

The spatiotemporal coincidence criteria are ±1°latitude, ±1°longitude and ±12 hours. They are similar to those used in Barret

et al. (2011), Boynard et al. (2016) (50 km ±10h), Boynard et al. (2018) (100 km, ±6h), Dufour et al. (2012) (110 km, ±7h). As we compare sondes with IASI morning data only and that most of the sonde launches are performed in the morning, using 6 or 12h coincidence does not introduce significant differences. We have computed statistics for 9 latitude bands which are the whole globe, the two hemispheres and six 30° wide latitude bands. For each band, the monthly mean is computed if there are more than 3 coincident profiles. Pixels are selected according to 3 quality criteria. We first keep pixels for which convergence

is achieved meaning a positive Jcost output from the 1DVar (based on gradient and evolution of Jcost between the two last iterations). We have also set an upper limit (1.0) for the retrieval cost in order to elliminate pixels with poor quality fits. Thirdly, only pixels with a total DFS > 2.0 are selected. Using these criteria we have kept about 9.0E5 pixels out of 1.1E6.

## 3.3 Comparison with raw and smoothed data

To compare remote sensed to in-situ or modeled profiles it is important to apply Eq. 1 to the in-situ or simulated profile (Rodgers, 2000; Barret et al., 2002). This procedure allows us to check the quality of the retrieval taking its degraded vertical resolution and sensitivity into account.

Nevertheless, in a validation objective it is also necessary to compare the retrieved profiles to raw (not smoothed by the AKs)

in-situ profiles in order to perform a fully informative validation. This is of particular importance when the satellite data are used for issues such as the ozone seasonal to interannual variabilities (Wespes et al., 2017; Peiro et al., 2018) or to document





the long term tropospheric ozone tendencies (Gaudel et al., 2018; Wespes et al., 2018; Dufour et al., 2018). Indeed, the application of Equation 1 implies the mixing of information between the different layers. Therefore, the variabilities and the drifts computed from raw and smoothed sonde data may be different and need to be documented. Raw ozone sonde data have been compared to IASI retrievals in few studies at the beginning of the IASI period (Barret et al., 2011; Dufour et al., 2012) but have

been disregarded in more recent validation work (Boynard et al., 2016, 2018). The importance of raw data validation regarding seasonal and interannual variabilities and trends analyses will be highlighted in details in section 4.

## 3.4   Taylor diagram

In order to validate remote sensing with reference in-situ observations we need to determine how well they are able to repro-

duce the same behaviour. There are four statistical indicators that have to be computed: (i) the absolute difference or bias which documents the accuracy, (ii) the root mean square of the differences (RMSD) which tells wether the bias is significant or not, (iii) the correlation coefficient (R) that document the consistency and phase of the variabilities of both datasets and (iv) the ratio of the standard deviations of both datasets which documents the goodness of the amplitude of the retrieval variability. In the case of IASI $O_3$, the first two indicators are frequently computed (Boynard et al., 2016, 2018; Barret et al., 2011) but the

last ones are rarely compared (Dufour et al., 2012) which makes most validation exercices incomplete.

Based on the relationship between correlation coefficients, RMSDs and variances of the reference (validating) and test (validated) datasets, Taylor has developed the Taylor diagram initially for climate models evaluation (Taylor, 2001). It displays all of these parameters (except the biases) in a more convenient and synthetic way than tables with numbers. Each experiment

or observation to be validated correspond to a point placed within a quarter circle. The reference is located in the middle of the X-axis (see Fig. 4, 5). The correlation coefficient between the reference and test dataset is given by the azimuthal position of the point. The RMSD is proportional to the distance between the test and the reference point. Finally, the radial distance from the origin is proportional to the variance of the experiment. Both RMSDs and standard deviations are normalised by the standard deviation of the reference (see Taylor (2001) for details). The normalisation allows us to display the results from

multiple experiments on a single diagram.

## 4   Validation results

### 4.1   General statistics for tropospheric, UTLS and stratospheric partial columns

For the different latitude bands, the statistics from the comparisons between ozonesondes and SOFRID data are presented in

Table 2 for the biases and corresponding RMSDs. Taylor diagrams are displayed in Fig. 4 for the TOC and lower tropospheric





columns and in Fig. 5 for the UTLS and stratospheric columns.

Concerning the troposphere, the comparison between SOFRID and raw sonde clearly shows the improvement from v1.6 to v3.5 (Fig. 4(a)). The v3.5 displays a larger variability in better agreement with the raw sonde data with a ratio between SOFRID

and sonde variances ranging from 0.62 to 1.01. For v1.6 this ratio ranges from 0.15 to 0.45. The RMSDs of the SOFRID versus raw sonde data are lower and the correlation coefficients larger for V3.5 than for V1.6. Tropospheric biases are smaller than 10% with the noticeable exception of mid and high latitudes of the SH for v1.6 and raw sonde data with significant biases of 29 and 55% respectively (Table 2). This problem of SOFRID v1.6 retrievals in the SH had already been diagnosed by Dufour et al. (2012) and by Emili et al. (2014). The use of a dynamical a priori in v3.5 allows us to reduce these large biases to almost

zero.

As expected, when the sonde profiles are smoothed with SOFRID AKs (Figure 4(b)) the agreement between sonde data and SOFRID retrievals is better. The retrieval variabilities are closer to the sonde variabilities, the RMSDs are smaller and the correlation coefficients are higher. It is also noticeable that differences between both retrieval versions is less important and

that the improvement of v3.5 relative to v1.6 is less evident. Furthermore, the large v1.6 biases in the SH troposphere at mid and high latitudes is reduced below 10 % when the impact of the a priori is taken into account with Equ. 1, hiding the problem.

The lower tropospheric retrieved columns agree less with raw sonde data with degraded correlation coefficients and larger RMSDs (Figure 4(c)) compared to the TOCs. For raw sonde data comparisons, the lower tropospheric variability is better for

V3.5 than for V1.6. When the sondes are smoothed, the statistics are much better and similar to the TOC results (Figure 4(d)). The added value of lower tropospehric columns relative to TOCs is therefore not obvious for SOFRID-O3.

In the UTLS, both v1.6 and v3.5 are in good agreement with raw sonde data (Figure 5(a)) and the differences between both versions are much lower than for the tropospheric columns. Correlation coefficients range from 0.67 to 0.93 and the ratios be-

tween retrieved and raw sonde variances range from 0.5 to 1.0 at mid and high latitudes. For the northern and southern tropical latitudes the correlation coefficients range from 0.6 to 0.75 and the variance ratios are between 1.6 and 2.1 highlighting a too high variability retrieved in the tropical UT. In the UTLS, biases are positive (5 to 18%) at high and mid latitudes and negative (-3 to -21%) at tropical latitudes and are not significant because of large RMSDs.

In the stratosphere, the agreement between raw sonde data and SOFRID retrievals is very good for the 2 versions and in all latitude bands with correlation coefficients in the 0.75-0.98 range and variance ratios in the 0.56-0.96 range except in the trop-ical bands where the retrieved variances are much lower than the ozonesonde variances (Figure 4(c)). Stratospheric columns from v3.5 are in slightly better agreement (higher r2, lower RMSDs) with ozonesonde data than v1.6. Large positive biases (10-14 %) are found at tropical latitudes for both v1.6 and v3.5 (Table 2).





Both in the UTLS and the stratosphere, the agreement is only slightly improved (larger correlation coefficients and lower RMSDs) when the sonde profiles are smoothed by the AKs (Figure 5(b) and (d)). Smoothing of the sonde profiles do not significantly modify the UTLS and stratospheric biases. In particular, the tropical UTLS large negative biases are still present when the AKs are applied to the sonde data. The small differences between v1.6 and v3.5 on the one hand and between raw

and smoothed sonde data on the other hand highlight the larger sensitivity of IASI to the UTLS and the stratosphere than to the troposphere as already discussed in Barret et al. (2011); Dufour et al. (2012) for SOFRID v1.5.

## 4.2   Vertical profiles

After comparing partial columns, it is interesting to look at complete profiles to get a better insight about the discrepancies

between IASI retrievals and sonde data. The annual average profiles for V1.6 and V3.5 are displayed in Fig. 6 and Fig. 7 resp. for the different latitude bands.

In the NH, v1.6 and v3.5 show similar behaviours with a large upper tropospheric positive bias at mid and high latitudes and a large oscillation from a negative bias at 250 hPa to a large positive bias at 100 hPa in the tropics. These profile features

are responsible for the positive (resp. negative) biases for the mid and high latitudes (resp. tropics) UTLS columns and for the positive biases for the tropical stratospheric columns (see Table 2). In the SH the large tropospheric positive biases of SOFRID relative to raw sondes (below 300 hPa in the high and mid-latitudes and below 500 hPa in the tropics) present in v1.6 almost dissapear in v3.5. The improvement of SOFRID accuracy in the SH extra tropical troposphere is the clearest advantage of using a dynamical a priori profiles. In the SH tropics, the TOC difference between v1.6 and 3.5 is not so clear (see Table 2) because

the positive bias in the lower troposphere is compensated by a larger negative bias in the upper troposphere in v1.6. As already discussed from column comparisons, it is also noticeable from profile comparisons (Fig. 6 and 7) that the agreement between SOFRID retrievals and smoothed sonde profiles is better than with raw sondes. An important exception is the large UTLS oscillations in both the NH and SH tropics and for both v1.6 and 3.5. Therefore, unlike expected, this important discrepancy between retrievals and sonde data does not result from the use of a single a priori profile too far from the real profile. The

differences between v3.5 and v1.6 are largely reduced when sondes are smoothed. For instance the large tropospheric biases for v1.6 in the SH disappears when the smoothing is applied to the sonde profiles.

For all latitude bands RMSD profiles display the largest values around the tropopause (below 60% in the extra tropics and up to 100% in the NH tropics) as is expected because it is the altitude range with the largest relative variability. RMS from

differences between retrievals and smoothed data are generally much lower than with raw data. This is also expected since the smoothing error is the largest source of error in IASI retrievals (see Barret et al. (2011); Dufour et al. (2012)). RMS of the differences with smoothed sondes in the troposphere are somewhat larger for v3.5 than v1.6 especially in the SH. This is an indication of the increased sensitivity and decreased smoothing of v3.5. This is also evident in the Taylor diagrams which show





that tropospheric variabilities are larger and in better agreement with sonde data (raw and smoothed) for v3.5 (see Fig. 4).

## 4.3  Time series of tropospheric columns

As tropospheric O$_3$ trends assessment is a major issue and one of the main topic of the TOAR (Tropopsheric Ozone Assessment

Report)/IGAC international initiative (Gaudel et al., 2018), we focus in this section on TOCs time series. Time series are also interesting to bring insight about the general statistics discussed in the previous sections and to identify possible drifts of the data.

The time series of IASI and sondes monthly TOCs are presented in Fig. 8 (resp. 9) for V1.6 and in Fig. 10 (resp. 11) for V3.5 for northern (resp. southern) hemisphere. We present both raw and smoothed sonde data to highlight the impact of smooth-

ing upon the agreement between IASI and sondes. This impact is particularly obvious for SOFRID v1.6 at mid-latitudes. At northern mid-latitudes the bias between SOFRID v1.6 and raw sonde TOCs displays large seasonal variations from -(5-10)% in summer to 10-20% in winter resulting in a negligible 2±15% average bias (Table 2). When sonde data are smoothed by IASI AKs, the sonde variability is largely reduced. Bias is varying from 5% in winter to -5% in summer.

For southern mid-latitudes, as already highlighted by Dufour et al. (2012) and Emili et al. (2014) SOFRID TOCs are significantly biased high (29±22%) relative to raw sonde data (Table 2). This was explained by the fact that the single a priori used in v1.6 is biased towards northern mid-latitude O$_3$ (Emili et al., 2014). When the sonde data are smoothed by IASI AKs, the agreement is much better and the bias becomes unsignificant (5±9%) as a result of taking the a priori contribution into account (Equ. 1). The largest significant bias (56±26%) is found in the SH high latitudes for v1.6 TOCs (Table 2) with large seasonal

variations from 20% in winter to 120% in summer. The large bias variabilities at mid- and especially high latitudes of the SH result from the very low seasonal variability of the retrieved columns (see Fig. 4(a)).

For V3.5, the use of a dynamical a priori profile clearly improves the retrievals at mid-latitudes. At northern mid-latitudes the seasonal bias variation is reduced to -10-0% and the average bias remains small (-6±14%). When smoothing is applied,

the seasonal variability almost disappears and the bias is only -3±9%. At southern mid-latitudes, the agreement is very good and very similar for raw and smoothed sonde data with no real seasonal signature detectable and an avergae bias close to 0%.

At tropical latitudes, the situation is quite different. First, the seasonal variability is not so notable and regular and the difference between raw and smoothed sondes is lower than at mid-latitudes. Furthermore, the behaviour of v1.6 and v3.5 are close

even though v3.5 is in better agreement with sonde data (see section 4.1). In the southern tropics there is a noticeable variation of bias between 2011-2014 with large negative biases of -10% and -15% and 2008-2010 and 2015-2017 with biases of 0 and -5% for v1.6 and v3.5 respectively. As such a bias variation is not detected for other latitude bands, we assume that it may be linked to a gap in sonde data for the 2011-2014 period. A closer look to SH tropics ECC sonde data show that only two stations (La Reunion and Nairobi) provide data regularly (30-50 profiles per year) over the period. For the Pago-Pago Pacific station



data are available only from 2014 to 2016 and since 2012 for Irene in South Africa. For the Natal Atlantic station more than 25 profiles are available during the 2008-2010 and 2014-2017 period and almost none during the 2011-2013 period.

One issue that was raised in TOAR (Gaudel et al., 2018) was the different trends computed from different satellite prod-
ucts. UV-Visible satellite sensors produce positive tropospheric $O_3$ burden trends in both hemisphere while trends from IASI products are negative. It has to be noted that in Gaudel et al. (2018), negative $O_3$ burden trends from SOFRID in the northern and southern hemisphere, and for the whole Earth are respectively 1/4, 1/2 and 1/3 smaller than FORLI's. The drifts computed from the SOFRID-sonde differences are displayed in Fig. 8 to 11.

At high northern latitudes, for both v1.6 and v3.5 the drifts are large (> 10 and >4.5%.decade$^{-1}$ for raw and smoothed data
resp.) and significant at the 95% level. For mid and tropical latitudes, drifts are between 0.9 and -3.4 %.decade$^{-1}$ but are not significant. The NH mid-latitude drift with raw sonde data is reduced from -3.2 with v1.6 to -0.6%.decade$^{-1}$ with v3.5. For the whole NH, the drifts are not significant and decreases from -2.2 with v1.6 to 0.7%.decade$^{-1}$ with v3.5 for raw sonde data. They are about 1.5$\pm$0.8%.decade$^{-1}$ and hardly significant (p>0.04) for smoothed sonde data.

In the SH tropics, drifts are ∼-5 and ∼-3%.decade$^{-1}$ for raw and smoothed sonde data resp. and only significant for v3.5 compared to raw data. These drifts are linked to the large negative biases of the 2011-2014 period resulting from misssing data (see above). For v1.6 a large but unsignificant drift (-8%) also occurs at high latitudes which is largely reduced for v3.5. For the whole SH we found a significant negative drift (relative to raw sonde data) of -9.5$\pm$4.7 for v1.6 which reduces to -4.3$\pm$1.4%.decade$^{-1}$ and becomes unsignificant for v3.5.

## 5   Comparison with IASI-FORLI

Two versions of IASI $O_3$ retrievals with the FORLI software have been validated by Boynard et al. (2016) (B16) and Boynard et al. (2018) (B18). Part of their validation results are based on the same data as the present study, namely ECC ozone sondes from the WOUDC database between 2008 and 2014 for B16 and 2008 and 2016 for B18. As they document the latest FORLI
version (v20151001) on a longer time period, we will focus our comparison with B18. They have used a comparable number (11600) of ozone sonde profiles than in the present study and their comparison methodology is close to the one we have used (spatio-temporal coincidence criteria set to 100 km and $\pm$ 6 h). We have collected the correlation coefficients (r2), the biases, the RMSDs from B18. We have also collected the DFS of the retrievals and the slopes of the linear fit between the smoothed sondes and retrievals from B18. There are two main limitations to the comparison between the validation of our SOFRID re-
trievals and the FORLI validation from B18. First, they do not document the sonde and IASI variabilities and make it therefore impossible to draw Taylor diagrams with their data. Second, they have limited their comparisons to smoothed sonde data and do not provide results from comparisons between raw sonde and FORLI data. In Fig. 14 we have drawn Biases and RMSDs from SOFRID v1.6 and 3.5 and from FORLI for the layers selected by B18 (1013-300 hPa, 300-150 hPa and 150-25 hPa) and



DFS in Fig. 12. Fig. 13 displays the correlation coefficients r2 and the slopes (b) from linear relationships fitted between IASI retrievals and smoothed sonde data. Finally, Fig. 15 document the drifts between sondes and SOFRID retrievals for the whole NH for the Surface-300 hPa layer to be comparable to B18.

In the three atmospheric layers, the information content is larger with FORLI than with SOFRID v1.6 and v3.5 (Fig. 12). This is particularly visible for the mid-latitudes and tropics in the troposphere with 0.8 to 0.9 DFS for FORLI and only 0.4 to 0.6 for SOFRID. This is probably a result from the choice of noise levels for both retrieval algorithms. At high latitudes the DFS are low and closer for both algorithms and the increase from high to mid-latitudes is therefore much larger for FORLI than for SOFRID. As both algorithms use a single retrieval noise and a priori covariance matrix and similar surface and atmo-
spheric temperatures, the reason for such a difference is unclear. In the UTLS and stratosphere the same increase of DFS from high latitudes to the tropics are visible for the three products. The difference in information content between retrievals is less pronounced in the UTLS and in the stratosphere than in the troposphere.

   The RMSDs (see Fig. 14) are generally larger for FORLI than for SOFRID. In the troposphere, RMSDs reach 18% for
FORLI and are below 10% for both SOFRID v1.6 and v3.5. In the UTLS, RMSDs are larger than in the other layers due to the lower absolute columns. For SOFRID UTLS RMSDs are in the range 10 to 30% and 20 to 45% for FORLI. For both SOFRID and FORLI the highest RMSDs are in the tropics where the 150-300 hPa columns are the lowest. In the stratosphere, FORLI's RMSDs are also systematically larger than SOFRID's. The differences are the largest at high latitudes with FORLI RMSDs 3 to 4 times larger than SOFRID's.
The r2 differences (Fig. 13) are partly related to the RMSDs differences. Generally, SOFRID has larger r2 than FORLI. As for the RMSDs, the differences between both algorithms are the largest at high latitudes (especially in the southern hemisphere where r2 < 0.4 for FORLI products) in the 3 layers. In the troposphere, the correlation coefficients are comparable for both algorithms in the tropical bands and SOFRID 3.5 give higher r2 than SOFRID 1.6. The differences between retrieval versions are generally lower and can even be reversed in the UTLS and in the stratosphere.

The slopes of the linear fits between retrievals and sonde data provide complementary information to the r2 coefficients. A slope smaller than one indicate than the retrieved variability is too low compared to the reference data and conversely, a slope larger than one indicate an overestimation of the variability. In the troposphere, SOFRID v1.6 and 3.5 and FORLI have similar slopes except in the 60-90S band where FORLI has a significantly lower slope than SOFRID (Fig. 13).

In the troposphere, FORLI products present systematic negative biases from 7 to 20% except in the polar regions. Concerning SOFRID, the tropospheric biases are within ±6% (comparable to TOCs biases in Table 2). The results are largely different when the raw sonde data are considered with very large biases in the southern hemisphere with SOFRID v1.6 as discussed in section 4.3. In the UTLS, SOFRID and FORLI biases are significantly positives except in the tropics and more specifically in
the SH tropics where SOFRID columns are negatively biased by ∼20% as discusssed in section 4.1 (Table 2). In the strato-





sphere, both SOFRID and FORLI products are positively biased. The largest differences between both retrieval algorithms are found in the extra tropical southern latitudes with FORLI biases larger than SOFRID. In the 60-90°S latitude band FORLI biases reach about 40 % against about 5% for SOFRID.

In the perspective of a better quantification of tropospheric $O_3$ evolution and of the TOAR results (Gaudel et al., 2018), it is also important to compare the drifts between sonde and retrievals. B18 present and discuss the drift between FORLI and sonde data for different layers in the whole NH. The SOFRID NH tropospheric drifts discussed in section 4.3 are smaller and opposite in sign to the significant -8.6±3.4%.decade$^{-1}$ drift between FORLI and smoothed sonde data in the NH troposphere presented in B18. As B18 computed a surface-300 hPa column instead of a tropospheric column, we have computed the drifts
based on the same layer (see Fig. 15). Drifts for Surface-300hPa columns are slightly (0.1 to 0.4%) smaller than for TOCs and are not significant in both cases. The comparison of the NH drift with B18 is therefore not dependent on the tropospheric layer definition. For v1.6 and v3.5 compared with raw and smoothed sonde data, the surface-300 hPa column drifts range from -2.0 to 1.3%.decade$^{-1}$ (see Fig. 15), values which are much smaller than in B18. These authors attribute their NH tropospheric significant drift to an abrupt change or jump detected in 2010 in FORLI but already detectable in the previous version
(v20141022) of FORLI (Boynard et al., 2016). No significant change occuring around 2010 is detectable for SOFRID v1.6 (Fig. 8(h)) and v3.5 (Fig. 10(h)) NH time serie. The difference could be linked to the use of EUMETSAT L2 products and of ECMWF analyses for FORLI and SOFRID retrievals respectively. As mentioned previously refering to B18, EUMETSAT L2 product are not homogeneous over the 2008-2016 period and a version change could result in the jump discussed in B18.

**6   Conclusions**

This study aimed at assessing the quality of two different versions of SOFRID-O3 at the global scale and over the 10 year IASI period using ozonesonde from the WOUDC. SOFRID-O3 v1.6 retrievals are based on a single a priori profile like most other global IASI $O_3$ retrievals (Barret et al., 2011; Dufour et al., 2012; Boynard et al., 2016, 2018). In V3.5 the a priori is dynamically selected from an $O_3$ profile climatology (Sofieva et al., 2014) based on latitude, season and the tropopause height.
Other satellite $O_3$ retrievals use a priori profiles from climatologies but they are chosen based on geographical and temporal criteria only (Bowman et al., 2006; Liu et al., 2010). To our knowledge it is the first time that the tropopause height is used for the choice of the a priori for spaceborne $O_3$ retrievals.

The general statistics (Taylor diagrams) of the comparisons between ozonesonde and SOFRID have highlighted the large
improvements brought by v3.5 especially in the troposphere. The use of a tropopause based a priori generally reduces the RMSDs and increases the r2 correlation coefficients and the amplitude of the retrieved variability. The high TOC biases of v1.6 relative to low $O_3$ is also corrected with v3.5. This is of particular importance in the SH extratropics where the very large



biases almost dissapear. In the NH lower TOC are retrieved in winter leading to a better seasonal cycle.

In the UTLS and stratosphere the improvements are less important. In particular both versions are impacted by positive biases for the UTLS (18% at NH mid-latitudes) and stratospheric (<7%) columns at extratropical latitudes that were already

discussed in Dufour et al. (2012). In the tropics large profile oscillations around the tropopause result in negative biases in the UTLS (21% in the SH) and positive biases (< 14%) in the stratospheric columns.

Concerning the TOC drifs, we have shown that there were no significant differences between v1.6 ans v3.5. There are no significant drifts except at high northern latitudes (increase of 9-13%.dec) and at southern tropical latitudes (decrease of 4-

5%.dec). For southern tropics, the apparent decrease is probably linked to a sampling weakness at different stations which makes the time serie inhomegeneous.

Our study have also demonstrated the importance of making comparisons with both raw and smoothed in-situ data. Comparing only with smoothed data could lead to the conclusion that the satellite data are better than they really are. For instance,

the high bias for low TOC with the v1.6 is almost completly corrected when smoothing is applied. The real improvement of v3.5 relative to v1.6 is only sizeable when we compare SOFRID retrievals with raw sonde data.

Finally we have compared our validation results to the latest (v20151001) FORLI-O3 retrievals validation. The comparison had to be limited because the variability of FORLI-O3 retrievals and ozonesonde data were not provided in Boynard et al.

(2018) which prevented us to draw Taylor diagrams. Furthermore, in Boynard et al. (2018) the FORLI-O3 are compared to smoothed sonde data only. FORLI produces larger RMSDs than SOFRID especially in the stratosphere at high latitudes. The correlation coefficients (r2) are consequently lower for FORLI columns than for SOFRID. Tropospheric biases are significantly larger for FORLI (7-20%) than for SOFRID (<6%). Finally, no significant tropospheric $O_3$ drift are detected for both versions of SOFRID-O3 in the NH. The difference with FORLI which is impacted by a significant -8.6%.decade$^{-1}$ drift are likely

linked to the use of different temperature profiles for the radiative transfer calculations (ECMWF analyses for SOFRID and EUMETSAT L2 for FORLI).

*Data availability.*   The SOFRID-O$_3$ data are freely available at the IASI-SOFRID website (http://thredds.sedoo.fr/iasi-sofrid-o3-co/).

*Author contributions.*   Brice Barret performed the validation of SOFRID-O$_3$ data and wrote the paper. Emanuele Emili initiated and con-

tributed to the development of SOFRID-O$_3$ v3.5. Eric Le Flochmoen is in charge of the SOFRID retrieval operations.



*Competing interests.* no competing interests are present for the present publication

*Acknowledgements.* IASI L1c data have been downloaded from the Ether French atmospheric database (http://ether.ipsl.jussieu.fr). The research with IASI is conducted with some financial support from the CNES (TOSCA–IASI project). The ozonesonde data used in this study were provided by the World Ozone and Ultraviolet Data Centre (WOUDC) (http://www.woudc.org). The authors thank those responsible for
5    the WOUDC measurements and archives for making the ozonesonde data available.





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





**Table 1.** Atmospheric layers for comparisons

| Layer | Lower boundary | Upper Boundary |
|---|---|---|
| Troposphere-1 | Ground | Tropopause |
| Troposphere-2 | Ground | 300 hPa |
| Lower Tropopshere | Ground | 550 hPa |
| UTLS | 300 hPa | 150 hPa |
| Stratosphere | 150 hPa | 25 hPa |

WMO: WMO: Meteorology - A three-dimensional science: Second session of the Commission for Aerology, WMO Bulletin, 4, 134–138, 1957.

Zhang, L., Li, Q. B., Murray, L. T., Luo, M., Liu, H., Jiang, J. H., Mao, Y., Chen, D., Gao, M., and Livesey, N.: A tropospheric ozone maximum over the equatorial Southern Indian Ocean, Atmospheric Chemistry and Physics, 12, 4279–4296, https://doi.org/10.5194/acp-12-4279-2012, 2012.

Ziemke, J., Oman, L., and Strode, S. e. a.: Trends in global tropospheric ozone inferred from a composite record of TOMS/OMI/MLS/OMPS satellite measurements and the MERRA-2 GMI simulation, Atmos. Chem. Phys., 19, 3257–3269, https://doi.org/10.5194/acp-19-3257-2019, 2019.



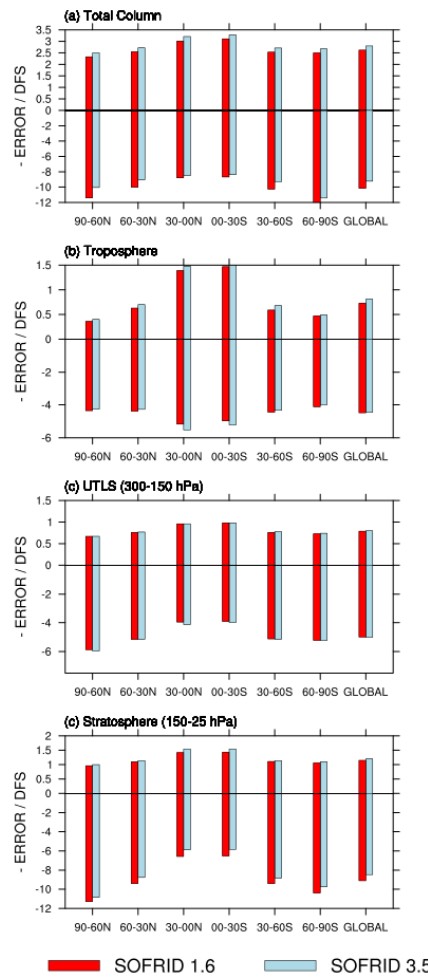

**Figure 1.** Degrees of Freedom for Signal (DFS) and retrieval errors for SOFRID-O3 V1.6 (red) and V3.5 (light blue) retrievals for (a) total column (b) Troposphere (c) UTLS (300-150 hPa) and (d) Stratosphere (150-25 hPa)





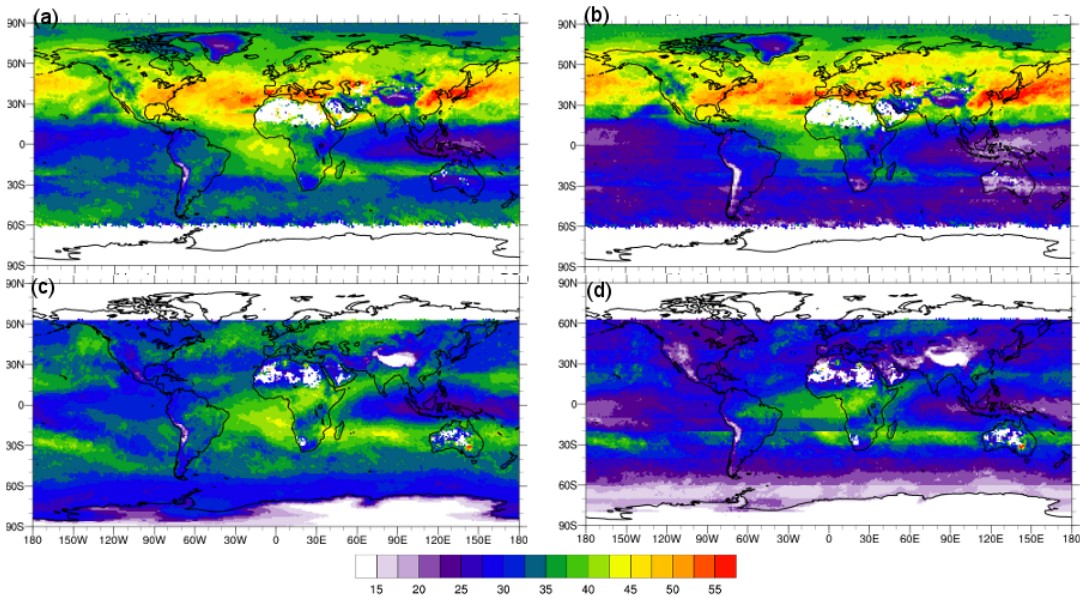

**Figure 2.** Tropospheric Ozone Column (TOC) distributions for (a) July 2017 v1.6 (b) July 2017 v3.5 (c) December 2017 v1.6 and (d) December 2017 v3.5.

**Table 2.** Biases between sondes and SOFRID retrievals with corresponding RMSDs. Values between brackets correspond to smoothed sonde data. Significant biases (Bias > RMSD) are in bold characters

| Latitude band | SOFRID | Troposphere | UTLS | Stratosphere |
|---|---|---|---|---|
| 90-60N | v1.6 | $6 \pm 14$ ($0 \pm 6$) | $6 \pm 18$ ($10 \pm 10$) | $7 \pm 10$ ($4 \pm 6$) |
|  | v3.5 | $-2 \pm 14$ ($-1 \pm 7$) | $10 \pm 15$ ($11 \pm 10$) | $1 \pm 3$ ($3 \pm 6$) |
| 60-30N | v1.6 | $2 \pm 15$ ($0 \pm 8$) | $18 \pm 27$ ($13 \pm 16$) | $2 \pm 8$ ($4 \pm 6$) |
|  | v3.5 | $-6 \pm 14$ ($-3 \pm 9$) | $17 \pm 27$ ($13 \pm 17$) | $1 \pm 7$ ($3 \pm 6$) |
| 30-00N | v1.6 | $2 \pm 17$ ($4 \pm 11$) | $-3 \pm 30$ ($1 \pm 37$) | **$14 \pm 8$ ($12 \pm 7$)** |
|  | v3.5 | $-3 \pm 16$ ($0 \pm 14$) | $-12 \pm 33$ ($-13 \pm 39$) | **$12 \pm 8$ ($12 \pm 7$)** |
| 00-30S | v1.6 | $-2 \pm 14$ ($-2 \pm 10$) | $-21 \pm 27$ ($-16 \pm 25$) | **$14 \pm 10$ ($10 \pm 8$)** |
|  | v3.5 | $-8 \pm 14$ ($-7 \pm 12$) | $-21 \pm 30$ ($-24 \pm 25$) | $10 \pm 11$ ($10 \pm 11$) |
| 30-60S | v1.6 | **$29 \pm 22$ ($5 \pm 9$)** | $11 \pm 29$ ($13 \pm 22$) | $1 \pm 8$ ($4 \pm 7$) |
|  | v3.5 | $1 \pm 18$ ($1 \pm 13$) | $10 \pm 28$ ($13 \pm 23$) | $3 \pm 7$ ($4 \pm 7$) |
| 60-90S | v1.6 | **$55 \pm 25$ ($7 \pm 6$)** | $5 \pm 22$ ($15 \pm 13$) | $7 \pm 12$ ($4 \pm 7$) |
|  | v3.5 | $0 \pm 16$ ($1 \pm 9$) | $7 \pm 19$ ($13 \pm 13$) | $6 \pm 11$ ($4 \pm 8$) |

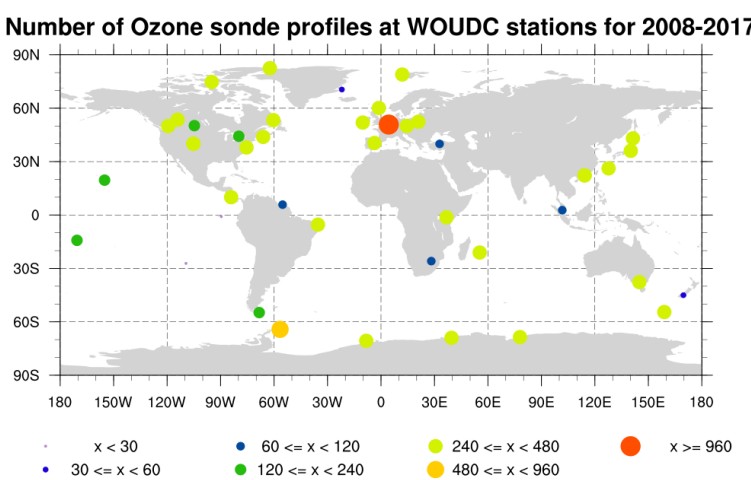

**Figure 3.** Maps of WOUDC stations with ECC O$_3$ sonde data during the 2008-2017 period. Colors and sizes of the markers indicate the number of valid sondes at each station.



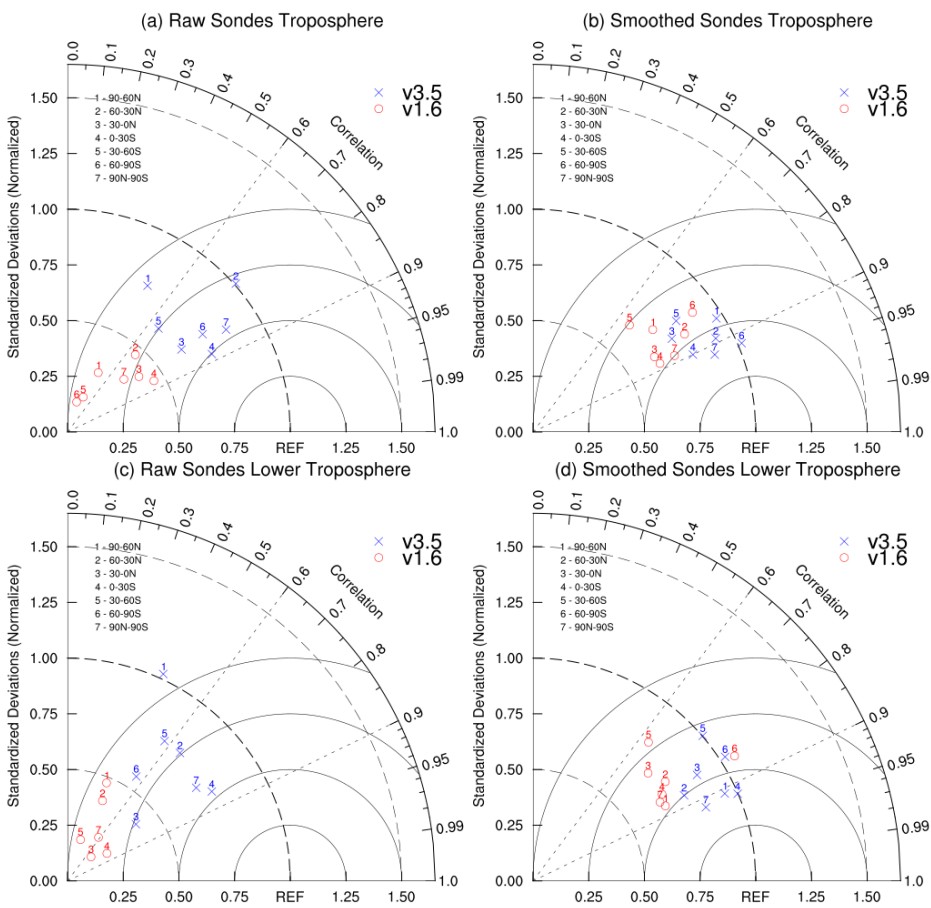

**Figure 4.** Taylor diagrams for (a) and (b) tropospheric columns and (c) and (d) lower tropospheric columns. (a) and (c) raw sonde data, (b) and (d) smoothed sonde data. Red circles (V1.6), Blue crosses (V3.5).



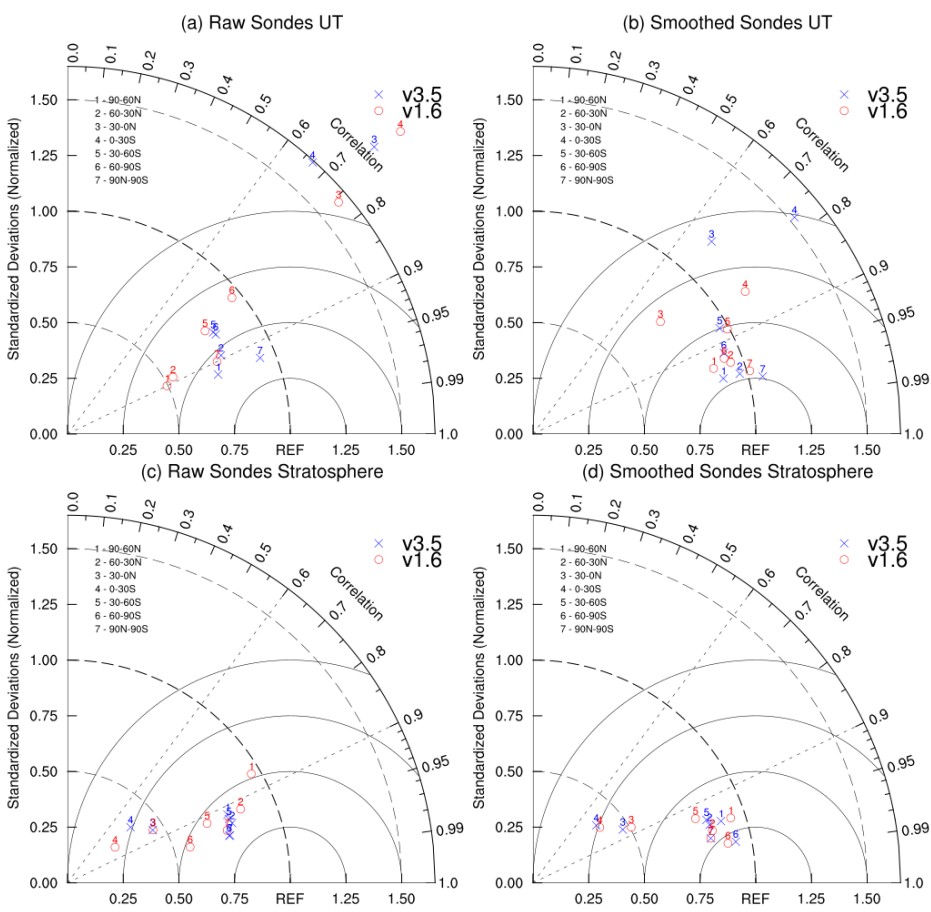

**Figure 5.** Taylor diagrams for (a) and (b) UTLS columns and (c) and (d) stratospheric columns. (a) and (c) raw sonde data, (b) and (d) smoothed sonde data. Red circles (V1.6), Blue crosses (V3.5).

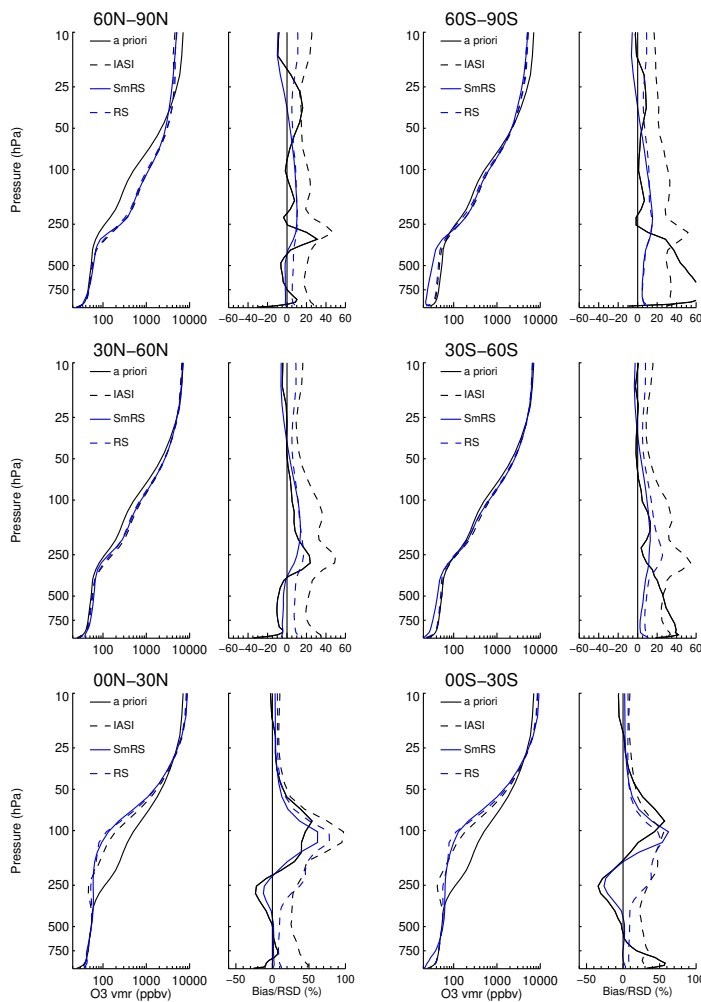

**Figure 6.** Profile comparisons between sonde and SOFRID-O3 v1.6 profiles (left panels) IASI, raw and smoothed sondes vertical profiles (right panels) biases (solid lines) and RMSD (dashed lines) between IASI and raw (black) and smoothed (blue) sondes for the NH (left panels) and SH (right panels).





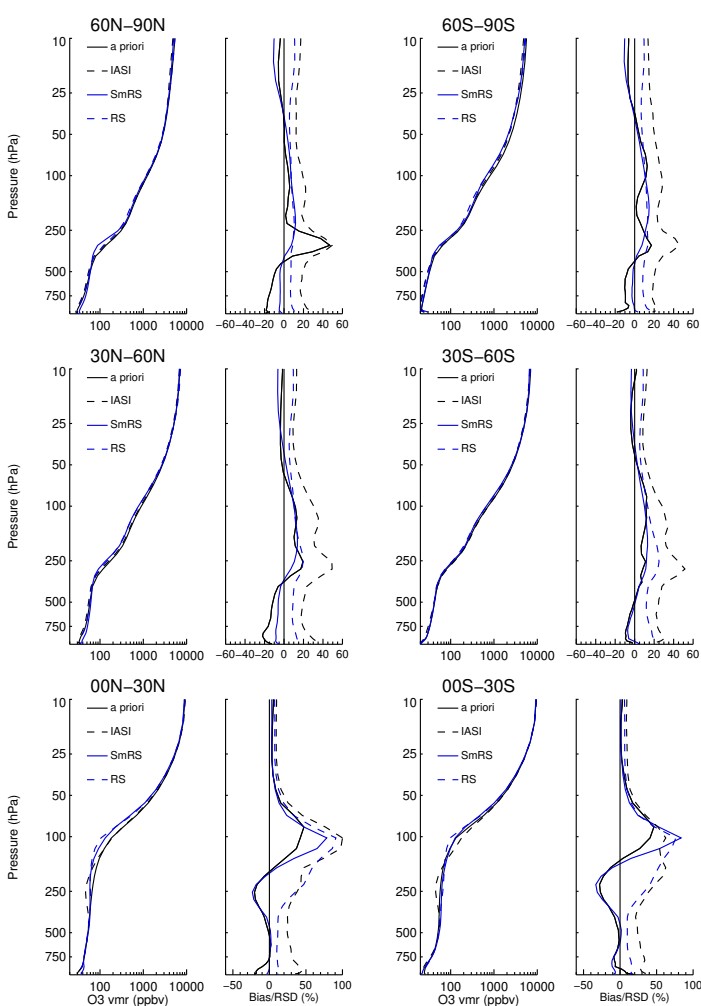

**Figure 7.** Same as 6 for SOFRID-O3 v3.5



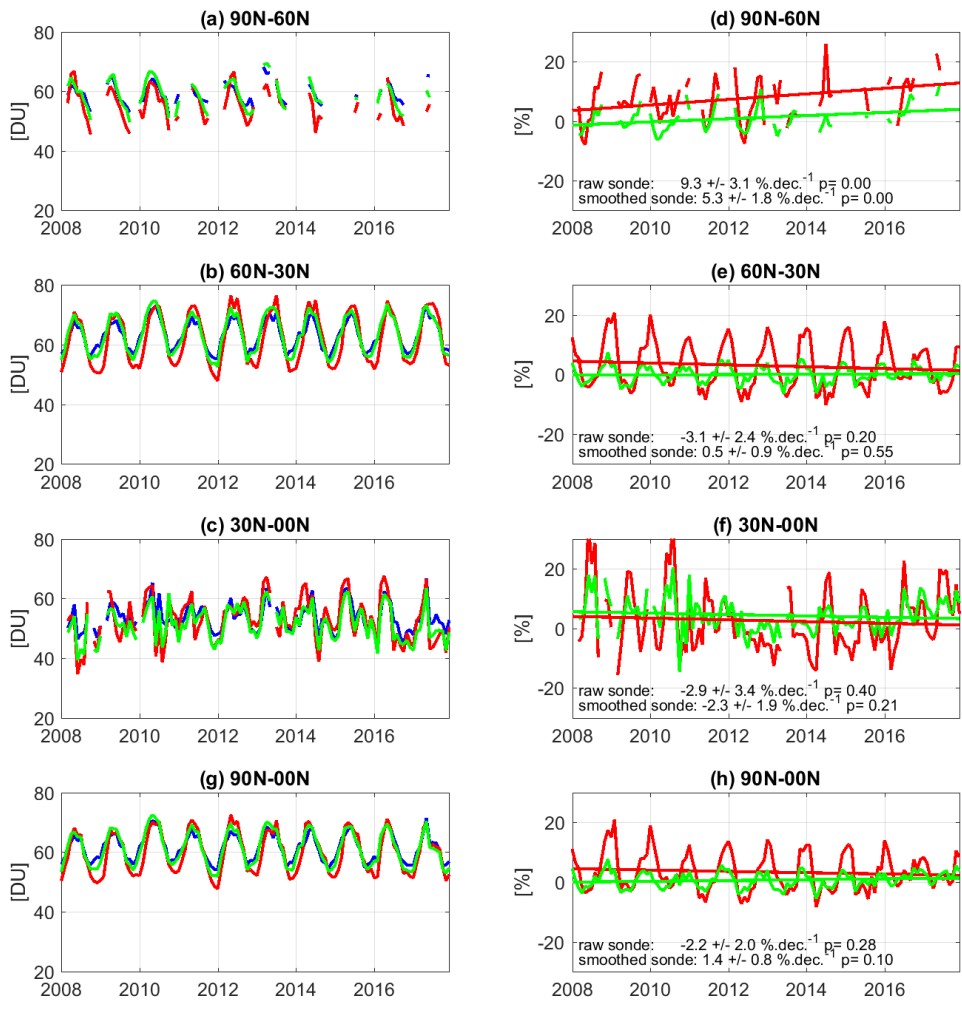

**Figure 8.** Time series of SOFRID-O3 v1.6 TOCs in the Northern Hemisphere for (a) 90-60°N (b) 60-30°N (c) 30-00°N (d) 90-00°N. Blue lines for IASI retrievals, red lines for raw sonde data and green lines for smoothed sonde data. Differences between IASI and sonde data for (e) 90-60°N (f) 60-30°N (g) 30-00°N (h) 90-00°N. Red lines for raw sonde data and green lines for smoothed sonde data.



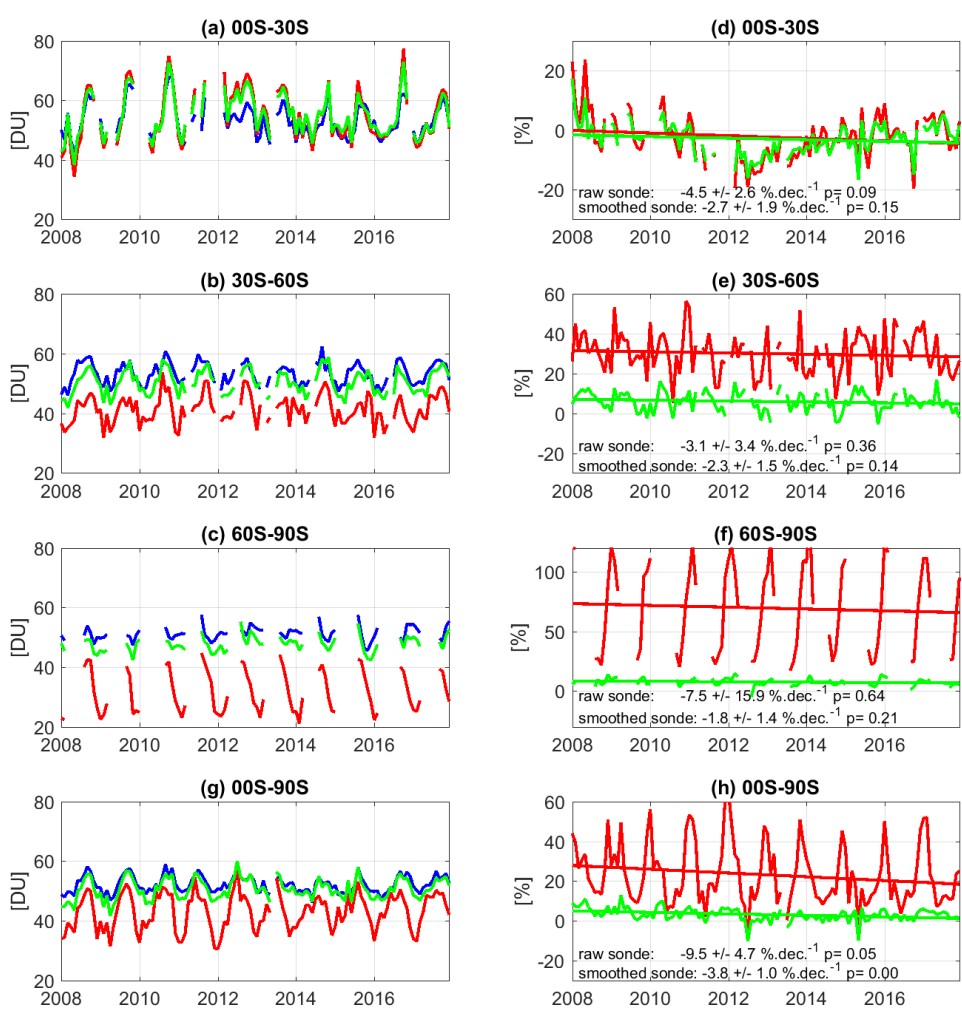

**Figure 9.** Same as figure 9 for SOFRID-O3 v1.6 in the Southern Hemisphere.



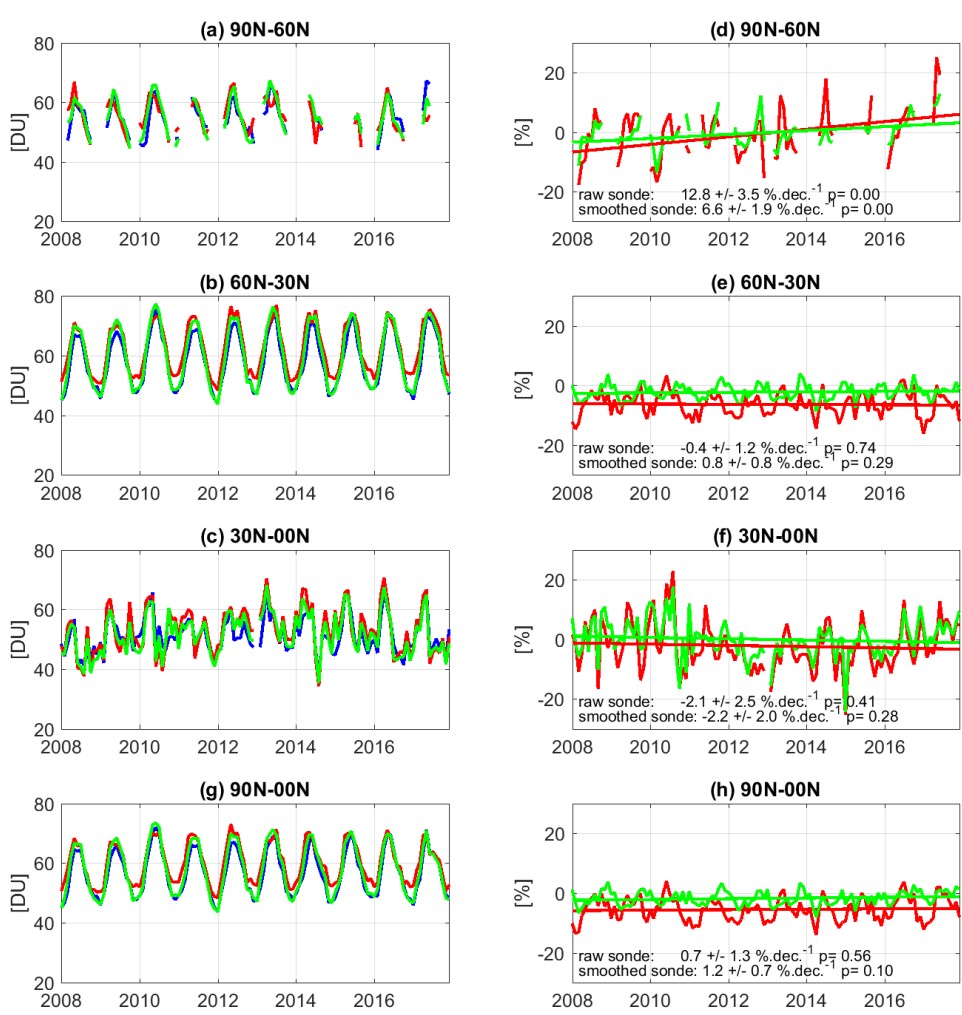

**Figure 10.** Same as figure 9 for SOFRID-O3 v3.5 in the Northern Hemisphere.



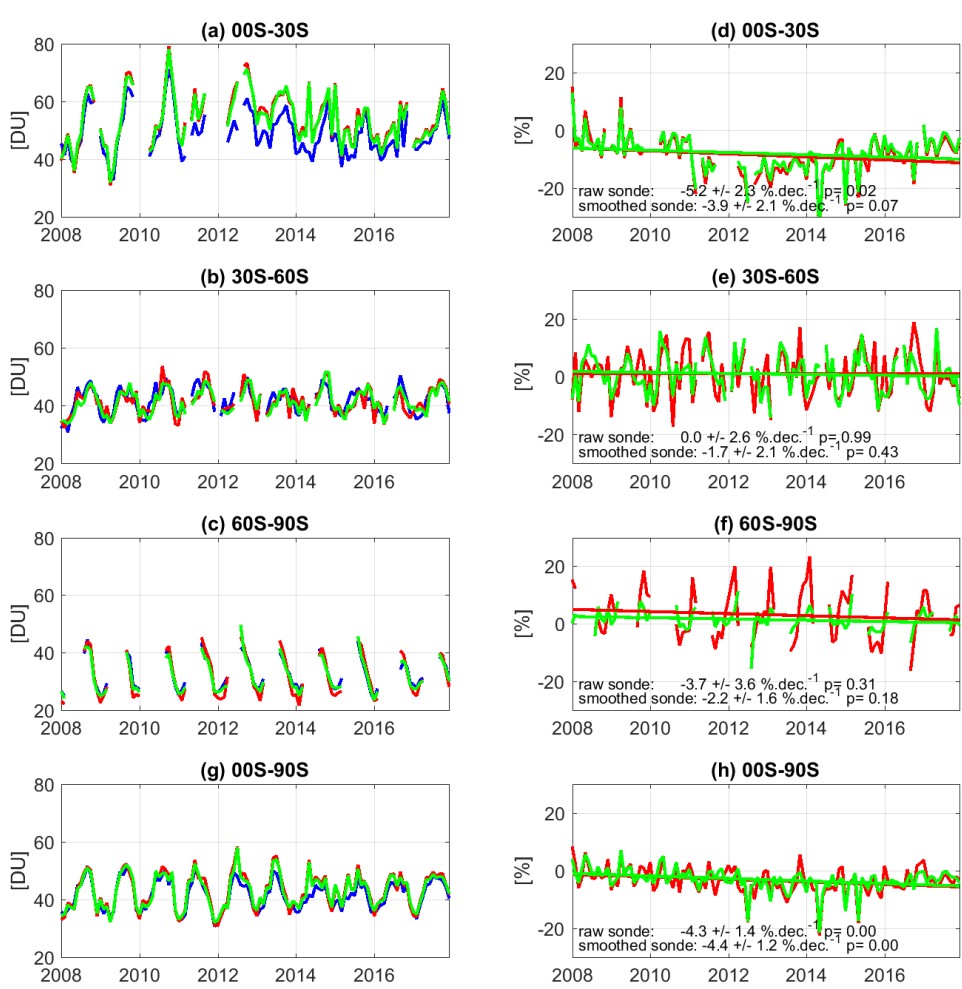

**Figure 11.** Same as figure 9 for SOFRID-O3 v3.5 in the Southern Hemisphere.

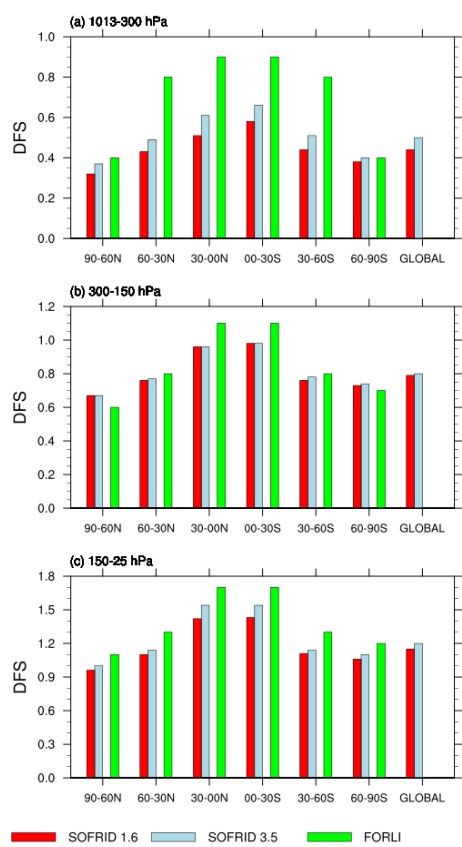

**Figure 12.** Degrees of Freedom for Signal (DFS) of IASI SOFRID-O3 v1.6 (red), SOFRID-O3 v3.5 (light blue) and FORLI O$_3$ retrievals in the different latitude bands for the (top) 1013-300 hPa (middle) 300-150 hPa and (bottom) 150-25 hPa.



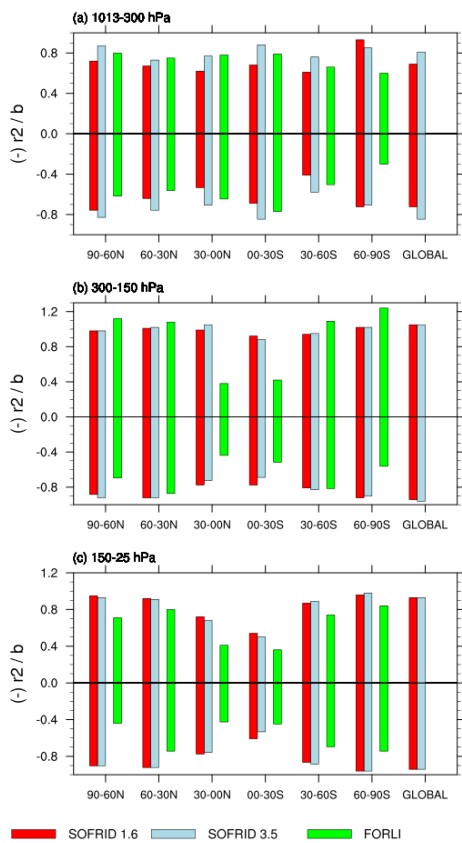

**Figure 13.** Slopes of the linear regression (positive values) and (-) r2 correlation coefficients (negative values) between IASI retrievals and sonde data.

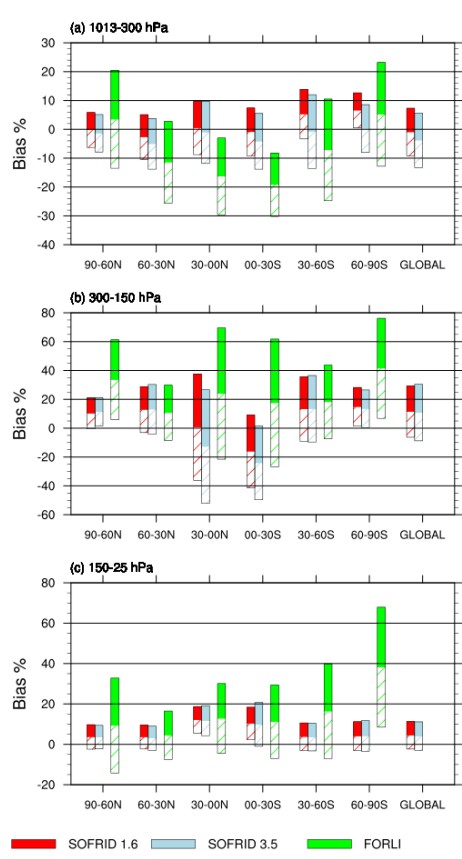

**Figure 14.** Biases and RMSDs of the differences between IASI retrievals and sonde data.



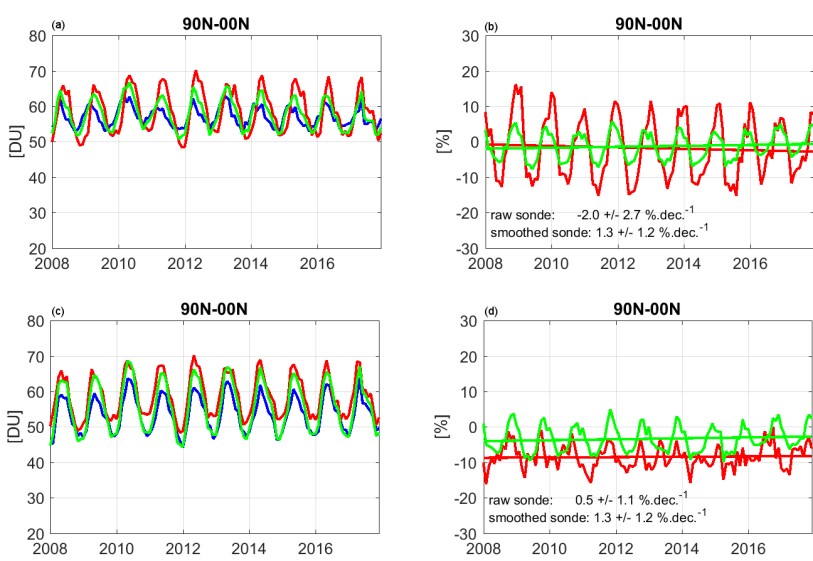

**Figure 15.** Time series of SOFRID-O3 (a) v1.6 and (c) v3.5 surface-300 hPa columns for the Northern Hemisphere (0-90°N). Blue lines for IASI retrievals, red lines for raw sonde data and green lines for smoothed sonde data. Differences between IASI and sonde data for (b) v1.6 and (d) v3.5. Red lines for raw sonde data and green lines for smoothed sonde data.