# Peer review of "A tropopause-related climatological a priori for IASI-SOFRID Ozone retrievals: improvements and validation"

_Atmospheric Measurement Techniques, 2020_

## Short Comment (SC1) · 29 Feb 2020

Dear Authors,

please note that a similar tropopause-based a priori selection, albeit less systematic than what you discuss in your manuscript, has been previously proposed and is discussed here: https://www.atmos-meas-tech.net/6/621/2013/amt-6-621-2013.pdf

See page 624: "The ozone a priori profiles used in the present work are derived from McPeters climatology (McPeters et al.,2007). To avoid numerical instability and aberrant oscillations in the retrieved profiles, we used a different a priori depending on

tropopause height derived from the pseudo-reality. We consider a tropopause higher than 14 km as a proxy for tropical air masses and then we used a tropical a priori (yearly climatological profile 20–30°N) in those cases. We used a mid-latitude a priori (summer climatological profile 30–60°N) in the other cases. The use of two different a priori, in particular mid-latitude and tropical climatological profiles, has been already successfully exploited for the LISA algorithm (Dufour et al.,2010)."

I suggest to cite our work as a previous example discussing the problem of using a single and immutable a priori.

Regards, Pasquale Sellitto
* * *

---

## Referee Comment (RC1) · Anonymous Referee #1 · 19 Mar 2020

The paper by Barret et al. presents a comparison of two versions of the IASI O3 product based on the SOFRID algorithm, one version (v1.6) using a single a priori profile, and the other version (v3.5) using a dynamical a priori based on climatological profiles depending on latitude, season and tropopause height. The paper also presents a comparison with another IASI O3 product using the FORLI algorithm. This comparison is based on FORLI's results from literature.

The paper is interesting mainly for two reasons: - the validation approach, including raw and smoothed profile and variability analysis, is very complete and provide guidelines for validation of satellite products. - The choice of the a priori profile to use in the

retrieval is still a question for the community. The paper shows the influence this choice can have on the retrieval results. For these reasons, the paper is suitable for publication in AMT. However, several key issues need to be addressed and better discussed before publication:

1. The new SOFRID product, v3.5, shows well visible stripes, especially in the southern hemisphere. Even if the authors mentioned that these discontinuities are consistent with their retrieval errors, these unphysical discontinuities would bring some difficulties to compare and evaluate models for example. The authors show how important it is to compare raw and smoothed data to satellite observation validation but it would be the same for model comparison and these stripes will compromise the comparison. The authors should mention these difficulties in using their product for model comparison and provide some possible ways to overcome these artifacts.

2. The authors state, especially in the conclusion, that the improvement of their new approach (dynamical a priori) is mainly due to the consideration of the tropopause height in the choice of the a priori. But they do not demonstrate the impact of taking an a priori profile on this basis, as they do not consider this selection independently from the latitude and season selection. Intuitively, we would expect that a better consideration of the tropopause would help resolving the biases in the UTLS, but no specific improvement are shown between v1.6 and v3.5. I would suspect that the bias correction is more related to the latitude/season selection as the v1.6 was based on a NH a priori only. The authors should better discuss this in the paper and show how the tropopause selection impact their retrieval if they think it is a key point. Moreover, the authors state that this dynamical approach based on the tropopause selection is presented for the first time. It is not completely true. The authors should refer to different publications using the KOPRAFIT-O3 algorithm they mentioned in the paper in which the selection of the a priori (and regularization) is based on the tropopause height (Dufour et al., ACP, 2015, 2018 and Eremenko et al., JQSRT, 2019).

3. Concerning the comparison with FORLI, as the authors use information from literature, the sampled pixels are likely different between the two algorithms and it can have a possible impact on the statistics, in particular if the cloud mask considered by the two algorithms is different. No information concerning the cloud filtering is mentioned for both SOFRID and FORLI. This should be added and discuss.

4. The quality of presentation of the results is sometimes poor and not suitable for publication. Figure and Table captions miss a lot of information such as units. A lot of typos remain. The authors should have read carefully their manuscript. In section 5 the authors referred to FORLI 16 and 18 on the Figures they comments but the Figures available online only show information named FORLI in green, whereas in the submitted paper for the quick review, both were present. Please, be consistent between the figure and the text.

Specific comments:

- p1, line 18: should we read theoretical or theoretically?

- P2, line 31: troposphEric

- P3, line24: quantify "weakly contaminated"

- P4, line 1 and 20: Is there a difference for ozone between HITRAN2004 and HITRAN2008?

- P5, line 11: emmissivity should be emissivity

- P5, Equation 1 : G is not define in the text.

- P5, line 22, change "devided" to "divided"

- P6, line 13: change "chinese" to "Chinese"

- P7, line 20: could you please clarify if the monthly mean is computed with at least 3 profiles by stations or by latitude bands? If it is by latitude bands, is it sufficiently representative?

- P7, line 21: Jcost is not defined

- P8, line 8: change "interanual" to "interannual"

- p8, line 13: documentS

- Table 2: please specify the units.

- P10, line 16: mid anD high latitudes

- P10, line 19: improvEment

- P10, line 27 and p11, line26: change "dissapears" to "disappears"

- P12, line 19: missing units

- P13, lines 10-13: the discussion is not clear. The authors state first the reason of the differences is the noise level and then it is not clear. They should provide the noise levels for the different algorithms to elaborate their hypothesis.

- P13, lines 6-34: the discussion is not consistent with the figure (no FORLI 16 and 18 display in the Figures).

- P14, line 5: change "positives" to "positive"

- P15, line 12: drifTs

- Figure 1: what are the error units? The way the authors present the plots with +/- for different variables is not very conventional. They should explain more precisely in the caption how to read the figure (this is also the case for Figs. 13 and 14).

- Figure 2: units are missing

- Figure 6: explain RS and SmRS, please

---

## Referee Comment (RC2) · Anonymous Referee #2 · 6 Apr 2020

The manuscript by Barret et al. presents the comparison and the validation of two versions of the SOFRID retrieval algorithm developed for IASI (SOFRID-O3 v1.6 and v3.5), which differ in apriori: single vs dynamical (month, latitude and tropopause height dependent). This study shows considerable work regarding the backprocessing of the whole IASI dataset with two recent versions of SOFRID, which use ECMWF operational analyses for temperature and humidity, in contrast with the previous versions. The comparison between products based on the use of a single vs variable apriori is particularly interesting, as the choice of the apriori remains an important source of discrepancy between retrieval datasets. The topic is suitable for ACP. However, I do have several key comments that should be addressed before publication:

[Figure]

General comments:

1/ My major comment is related to the validation methodology used by the authors; I do have doubts about the fact that ozonesonde data are used both for building apriori (single and variable) and for the validation. It is commonly accepted that one specific dataset or instrument cannot be used both for the apriori used for the retrieval and for the validation of the corresponding retrieved product, for evident reasons. Even if the IASI period validated here (2008-2017) is different from that used to build the apriori (1980-2006 for V3.5 and 2008-2009 for V1.6, hence, the WOUDC measurements used to generate the V1.6 single apriori are included in the validation dataset), I'm wandering to what extent it might affect the results. The a priori contribution contained in the retrieved product would tend to improve the comparison. That a priori contribution can be easily calculated and should be discussed in the validation section. Please discuss that point.

2/ Section 3.3: Even if not necessary for a pure validation exercise, the comparison with raw vs smoothed data is interesting as it allows a better evaluation of the O3 variability captured by the instrument. However, one should note that when considering variable apriori (according to season, location and tropopause height), a part of the expected variability is injected by default in the retrieved product through the a priori contribution, making the comparison with raw data wrongly improved when using variable vs single apriori. In addition, the presence of visible stripes (Section 2.4) due to the use of variable apriori that depend on location may constitute an issue for further comparison study, e.g. with CTM. This is exactly why, one can usually prefer using a single vs variable apriori profiles; it gives a homogeneous retrieval at the global scale and the retrieved variability is not distorted by that of the variable apriori. Hence, the true capability of the pair instrument/algorithm to capture the O3 variability is better infer when using a single apriori profile. That point should be clearly discussed in Sections 3.3 and 4.

3/ Through Section 4, the authors insists on the fact that "the improvement of SOFRID

accuracy . . . is the clearest advantage of using a dynamical apriori profiles". Given that several sources of improvement are taken into account: dependence on tropopause height, latitude and month, how can the authors be able to dissociate between their respective effects? Please, provide sensitivity tests or clarify that point?

Comments 2/ and 3/ highlight the limitations in using variable apriori and evaluating the V3.5 product. The authors should better discuss those issues through the manuscript in order to get a better feeling for the real advantage of using variable apriori (in terms of both location,season and dynamical tropopause).

4/ Regarding the comparison with FORLI, the authors are very negative through the manuscript and the critics are most of the time out of context. For instance:

- In the abstract: "(iii) in the N.H., no significant temporal drift is detected in SOFRID contrarily to FORLI ($\sim$8%)"

- Introduction, L21: "They both document a problem (drift or jump) . . ."

- Section 5, p.14, L.7-9: "the SOFRID NH tropospheric drifts discussed in section 4.3 are smaller and opposite in sign to the significant -8.6$\pm$3.4%/dec drift between FORLI and smoothed sonde data in the NH troposphere presented in B18."

That comparison of the "drift" calculated from SOFRID vs FORLI does not make sense. Indeed, the authors have to make a clear distinction between a "drift" that usually refers to an instrumental drift in validation studies, and a "jump" (or sudden discontinuity) as observed in the FORLI dataset, which induces an artificial drift, in order to avoid any confusion. It has already been clearly explained and discussed in Boynard et al. (2018) and in Wespes et al. (2018; 2019): the drift strongly decreases (< 1DU/dec on average) after the jump and it becomes even non-significant for most of the stations over the periods before or after the jump, separately. The discontinuity is strongly suspected to result from updates in level-2 temperature data from Eumetsat, which occur at the same date of the detected jump and which are used as inputs into FORLI. Hence, it

is obvious that "No significant change occuring around 2010 is detectable for SOFRID v1.6 (Fig. 8(h)) and v3.5 (Fig. 10(h)) NH time series", given that SOFRID uses L2 from ECMWF, not from EUMETSAT. It should be clarified through the manuscript.

- Section 4.3, p.12, L.6-7: It has also to be clearly noted that Gaudel et al. (2018) study suffers from a lack of consistency between a series of parameters, such as the calculation of the tropopause, making the comparison not quantitative.

- Section 5, p.12, L.32-33: First of all, on the contrary to what is stated in Section 3.4, three indicators (not only two) were calculated in Boynard et al. (2016, 2018), the fourth one (ratio of std) being rarely calculated in validation studies. That last one that makes possible to draw Taylor diagram is indeed interesting as it allows evaluating the representation of the retrieved variability. It could indeed be investigated for the validation of future FORLI products. Nevertheless, I am surprised that the authors did not perform their own analysis using the FORLI dataset that is publicly available on the french Ether/Aeris platform. It would have prevented possible inconsistencies between the SOFRID and the IASI datasets, the validation methodologies... For instance, in:

- Section 5, p.13, L9-10: One source of difference between FORLI and SOFRID could be the series of quality flags that have been applied on the datasets to select the best observations in terms of spectral fit and cloudy scenes. Are the flags comparable between the FORLI and the SOFRID datasets? Please comment.

This is why taking data directly from literature for a quantitative comparison might be inappropriate and mislead the comparison. That issue/limitation in the comparison between SOFRID and FORLI should be clearly highlighted and discussed by the authors. I would strongly recommend the authors to better put the FORLI-SOFRID comparison into context with the reasons mentioned here above (i.e. jump in contrast with real drift, use of different quality flags, possible inconsistency between validation methodology...) through the manuscript.

Minor comments:

[Figure]

- P.6, L.6-7: Why the behavior of TOC errors is similar to that of DFS while one can read above that the dominant source is the smoothing error? Please explain.

- P.10, L.2-3: Why does the smoothing of sonde profiles not improve the bias in UTLS while the DFS is < 1? Please explain.

- Regarding the figures 12-14, one could think that the authors make their own analysis from the FORLI datasets, while the values are taken from previous validation papers. This should be clearly mentioned in the figure captions to avoid misunderstandings.

Technical comments and typos:

- P.2, l.22: The jump is detected in year 2010, not 2011.

- P.2, L.30: tropospheric -> tropospheric

- P.3, L.7: methodology -> methodology

- P.4, L.33: "The use OF a . . ."

- P.5, L.8: atmospheric -> atmospheric

- P.6, L.1: Th -> The

- P.7, L.20: one reference is missing here.

- P.7, L.9: below -> above

- P.7, L.21: elliminate -> eliminate

- P.8, L.23: variance -> ratio of the variance (?)

- Table 2: Units are missing

- P.9, L.21: tropospehric -> tropospheric

- P.9, L.27: UT -> UTLS

- Fig.6 and 7: The legend is not clear. I guess RS means Raw Sondes and SmRS

means Smoothed Sondes. Hence, SmRS should be SmS (?). Please correct or clarify in the caption.

- Error in the caption of Fig.9: "Same as Fig.9" -> "Same as Fig.8"

- Fig.8: The color legend should be indicated in the top panels.

- P.12, L.6: Which version of SOFRID are you referring to?

- Fig.12 to 14 do not seem in correct order. Please consider this:

Fig.14 -> Fig.12, Fig.12 -> Fig.13, Fig.13 -> Fig.14

- P.13, L.1: delete "(b)" in the sentence. I don't see that in Fig.13.

---

## Author Comment (AC1) · 22 Jun 2020

We have acknowledge the work of Sellito et al. (2013) concerning the use of two different a priori profiles based on tropopause height in the revised manuscript.

---

## Author Comment (AC2) · 24 Jun 2020

**Answers to reviewer #1 :**

*1. The new SOFRID product, v3.5, shows well visible stripes, especially in the southern hemisphere. Even if the authors mentioned that these discontinuities are consistent with their retrieval errors, these unphysical discontinuities would bring some difficulties to compare and evaluate models for example. The authors show how important it is to compare raw and smoothed data to satellite observation validation but it would be the same for model comparison and these stripes will compromise the comparison. The authors should mention these difficulties in using their product for model comparison and provide some possible ways to overcome these artifacts.*

Please see our reply to reviewer #2 to a similar comment (comment #2).

*2. The authors state, especially in the conclusion, that the improvement of their new approach (dynamical a priori) is mainly due to the consideration of the tropopause height in the choice of the a priori. But they do not demonstrate the impact of taking an a priori profile on this basis, as they do not consider this selection independently from the latitude and season selection. Intuitively, we would expect that a better consideration of the tropopause would help resolving the biases in the UTLS, but no specific improvement are shown between v1.6 and v3.5. I would suspect that the bias correction is more related to the latitude/season selection as the v1.6 was based on a NH a priori only. The authors should better discuss this in the paper and show how the tropopause selection impact their retrieval if they think it is a key point.*

The same point regarding the relative importance of the tropopause-, latitude- and month-dependence of the a priori has ben raised by reviewer #2. The important point is that tropopause and « latitude/season» are strongly correlated and it is therefore not possible to fully adress this question. The Sofieva et al. (2014) climatology provides an information about the intra-seasonal tropopause related variability of the O3-profile on top of the larger seasonal variability. We have performed a sensitivity test to separate the impact of the intra-seasonal tropopause-dependence from the seasonal variability. The results are given and discussed in our answer to reviewer #2 (comment #3).

Concerning specifically this comment of reviewer #1: it is true that the UTLS bias has not been corrected by the dynamical a priori which is an important result: our papers further demonstrates that this bias is not or little related to the a priori as highlighted in the paper. Nevertheless, the UTLS variability has been largely improved with v3.5 (see Taylor diagram Fig. 5a) showing the advantage of a climatological a priori for UTLS retrievals. As discussed in the answer to reviewer # 2, part of this improvement is due to the intra-seasonal variability of the a priori.

*Moreover, the authors state that this dynamical approach based on the tropopause selection is presented for the first time. It is not completely true. The authors should refer to different publications using the KOPRAFIT-O3 algorithm they mentioned in the paper in which the selection of the a priori (and regularization) is based on the tropopause height (Dufour et al., ACP, 2015, 2018 and Eremenko et al., JQSRT, 2019).*

We acknowledge the work of Sellito et al. (2013) (reply to public comment by Pasquale Sellito in this discussion), Dufour et al. (2015) and Eremenko (2019) with KOPRAFIT-O3 in the manuscript. The approach by Sellito (2013) and Dufour (2015) are similar and rather basic using only 2 and 3 different a priori profiles for tropical and mid latitudes and for high, mid and tropical latitudes respectively. Eremenko et al. (2019) developped a more sophisticated method that has only been tested on synthetic data and not on real satellite observatioons. We have therefore added the following statements:

- In section 2.2: «… In a first attempt to take this tropopause effect into account for satellite data, Sellito et al. (2013) have implemented 2 a priori profiles in the KOPRAFIT-O3 retrieval algorithm to basically discriminate tropical  (tropopause higher than 14 km) from other latitudes. Dufour et al. (2015) have slightly improved the approach with a set of 3 a priori profiles for high latitudes (tropopause lower than 10km),  mid-latitudes (tropopause between 10 and 14 km) and the tropics (tropopause higher than 14 km). Eremenko et al. (2019) have tested a set of N profiles for retrievals on a synthetic database.»

- In the conclusion: «… Other satellite O3 retrievals use a priori profiles from climatologies but they are chosen based on geographical and temporal criteria only (Bowman et al., 2006; Liu et al., 2010). Dufour et al. (2015) use 3 different a priori profiles picked up according to 3 broad tropopause height classes to represent high, mid and tropical latitudes...»

In the present paper we use hundreds of a priori profiles based on latitude (10° bins), month and tropopause (1 km bins). It is therefore really the first time that such a *comprehensive* approach is used for real data retrievals. Therefore:

- in the abstract we keep our statement just adding the word *comprehensive* to differentiate our approach from Dufour et al. (2015) : «For the first time we have implemented a *comprehensive* dynamical a priori profile for spaceborne O3 retrievals which takes the pixel  location, time and tropopause height into account for SOFRID-O3 v3.5 retrievals.»

- similarly, in the conclusion we have added *in such a comprehensive way* : « To our knowledge it is the first time that the tropopause height is used *in such a comprehensive way* for the choice of the a priori for spaceborne O3 retrievals.»

*3. Concerning the comparison with FORLI, as the authors use information from literature, the sampled pixels are likely different between the two algorithms and it can have a possible impact on the statistics, in particular if the cloud mask considered by the two algorithms is different. No information concerning the cloud filtering is mentioned for both SOFRID and FORLI. This should be added and discuss.*

The same issue was raised by reviewer #2.
We have added missing information concerning our cloud mask and we have made sensitivity tests with  different values of our data filters such as cloud mask, cost function and DFS. In each case, the general statistics on which the FORLI-SOFRID comparisons are based are not altered.
See our detailed reply to reviewer #2 comment #4 « *Section 5, p.13, L9-10* ».

*4. The quality of presentation of the results is sometimes poor and not suitable for publication. Figure and Table captions miss a lot of information such as units. A lot of typos remain. The authors should have read carefully their manuscript. In section 5 the authors referred to FORLI 16 and 18 on the Figures they comments but the Figures available online only show information named FORLI in green, whereas in the submitted paper for the quick review, both were present. Please, be consistent between the figure and the text.*

We have completed the captions of the Figures and Tables with missing information such as units. We have also improved other presentation details mentioned by the reviewer. We have removed remaining mention to FORLI 16 and 18 as we have just kept comparisons with the latest FORLI-O3 version.

*Specific comments:*
*- p1, line 18: should we read theoretical or theoretically?*

Theoretical as the information is theoretically computed.

*- P2, line 31: troposphEric*

OK

*- P3, line24: quantify "weakly contaminated"*

We have added our AVHRR cloud fraction cover threshold in the manuscript : 25 %.

*- P4, line 1 and 20: Is there a difference for ozone between HITRAN2004 and HI-TRAN2008?*

A major update concerning line positions, intensities and lower state energy has been made for the three main O3 isotopologues in HITRAN2008 according to Rothman et al. (2009). Nevertheless, this update marginally concerns the 9.6 microns absorption band used for IASI O3 retrievals. We therefore think that it does not affect significantly SOFRID O3 retrievals.

*- P5, line 11: emmissivity should be emissivity*

OK

*- P5, Equation 1 : G is not define in the text.*

We have added its definition : « G is the gain matrix that represents the sensitivity of the retrieval to the measurement. »

*- P5, line 22, change "devided" to "divided"*

OK

*- P6, line 13: change "chinese" to "Chinese"*

OK

*- P7, line 20: could you please clarify if the monthly mean is computed with at least*
*3 profiles by stations or by latitude bands? If it is by latitude bands, is it sufficiently*
*representative?*

In fact we use at least 4 profiles (more than or $> 3$ ) per latitude band. We have corrected in the paper :

« … at least 4 coincident profiles within this latitude band »

We have performed tests with higher and lower numbers of sondes required per month and latitude band. We found that 4 was a good compromise between a representative monthly O3 in a 30° band and two many lacking months in the time serie. Requiring more profiles mostly impacted the time series in the southern tropical band which is rather poorly sampled. We discuss this issue «only two stations (La Reunion and Nairobi) provide data regularly (30-50 profiles per year) over the period» and its implications (bias variability over the period) in the paper.

*- P7, line 21: Jcost is not defined*

We have added its definition « retrieval cost function ».

*- P8, line 8: change "interanual" to "interannual"*

OK

*- p8, line 13: documentS*

OK

*- Table 2: please specify the units.*

OK

*- P10, line 16: mid anD high latitudes*

OK

*- P10, line 19: improvEment*

OK

*- P10, line 27 and p11, line26: change "dissapears" to "disappears"*

ok

*- P12, line 19: missing units*

OK

*- P13, lines 10-13: the discussion is not clear. The authors state first the reason of the*
*differences is the noise level and then it is not clear. They should provide the noise*
*levels for the different algorithms to elaborate their hypothesis.*

We have added a ref to Dufour et al. (2012) where the noise levels are given for both algorithm. This is enough for the qualitative argument concerning the DFS given here.

« This probably results from the retrieval noise level which is lower for FORLI than for SOFRID (Dufour et al., 2012) »

*- P13, lines 6-34: the discussion is not consistent with the figure (no FORLI 16 and 18 display in the Figures).*

OK

*- P14, line 5: change "positives" to "positive"*

OK

*- P15, line 12: drifTs*

OK

*- Figure 1: what are the error units? The way the authors present the plots with +/- for different variables is not very conventional. They should explain more precisely in the caption how to read the figure (this is also the case for Figs. 13 and 14).*

The units have been added. The errors are given as negative values with « -ERROR » indicated on the y axis. This way of presenting allows to plot two variables at the same time.

We have added (-) in the caption for things to be clearer.

*- Figure 2: units are missing*

Units « Dobson Units (DU) » has been added.

*- Figure 6: explain RS and SmRS, please*

The captions have been updated.

---

## Author Comment (AC3) · 24 Jun 2020

**Answers to reviewer #2 :**

*1/ My major comment is related to the validation methodology used by the authors; Ido have doubts about the fact that ozonesonde data are used both for building apriori (single and variable) and for the validation. It is commonly accepted that one specific dataset or instrument cannot be used both for the apriori used for the retrieval and for the validation of the corresponding retrieved product, for evident reasons. Even if the IASI period validated here (2008-2017) is different from that used to build the apriori (1980-2006 for V3.5 and 2008-2009 for V1.6, hence, the WOUDC measurements used to generate the V1.6 single apriori are included in the validation dataset), I'm wandering to what extent it might affect the results.*

We do not agree with the reviewer about the fact that the use of O3 profiles from the WOUDC ozonesonde dataset for both the a priori and the validation could cause a problem in the validation methodology. Moreover we are not aware of publications which would have highlighted this issue. We have mostly two objections about the reviewer statement #1:

- First and most importantly, the a priori for an OEM algorithm is not equivalent to a training dataset for an AI (Artificial Intelligence) or a NN (Neural Network) retrieval algorithm or to the ensemble used to constrain a model to provide analyses with a data assimilation systems. For an AI or NN algorithm, the retrieved quantities are strongly bounded within the variability of their training datasets and for an assimilation system the analyses will likely provide better comparisons if compared with assimilated data. For an OEM algorithm, the a priori is built from an ensemble of data (mixture of ozone sondes and satellite datasets in our case) to provide the best knowledge of the **average** state and of its **variability**. The ozone sonde profiles from the ensemble are not used individually to train or constrain the algorithm. Therefore we can consider that our a priori data and each sonde profiles are completely independent.

- Second, the O3 sonde instruments are state of the art calibrated and validated in situ instruments. They provide O3 concentrations as close as possible to the « truth ». Each sonde can reasonably be considered as independent from each other. They are very different from remote sensing measurements with limited vertical sensitivity and likely systematic biases. Using observations from a satellite instrument to build an a priori and validate the instrument using this a priori with the same data would raise issues related to the reviewer concerns.

*The a priori contribution contained in the retrieved product would tend to improve the comparison. That a priori contribution can be easily calculated and should be discussed in the validation section. Please discuss that point.*

The a priori is used to complete the information provided by the instrument in the part of the state vector space (O3 profile in our case) where the instrument is not providing information (see Rodgers 2000 for instance). Understandably, this information has to be as accurate as possible. I will rather return the question:

why should we use a single a priori when there are comprehensive and state of the art climatologies available which give better results ?

See reply to next comment which raises the same concern. The quote of Rodgers (2000) (who theorised the OEM for atmospheric soundings) advocates for a climatological latitude and time dependent a priori.

The a priori contribution is theoretically calculated and provided in the paper as the smoothing error based on the retrieval equation (Eq. 1 and Fig. 1 in the manuscript).

Furtjhermore, the paper gives a rather complete evaluation of the a priori «contribution» on the retrieval comparing retrievals from V1.6 and V3.5. The conclusion is that using a good a priori significantly improves rather than « *tends to improve* » the retrievals based on comprehensive comparisons with 10 years of global ozone soundings.

*2/ Section 3.3: Even if not necessary for a pure validation exercise, the comparison with raw vs smoothed data is interesting as it allows a better evaluation of the O3 variability captured by the instrument. However, one should note that when considering variable apriori (according to season, location and tropopause height), a part of the expected variability is injected by default in the retrieved product through the a priori contribution, making the comparison with raw data wrongly improved when using variable vs single apriori. In addition, the presence of visible stripes (Section 2.4) due to the use of variable apriori that depend on location may constitute an issue for further comparison study, e.g. with CTM. This is exactly why, one can usually prefer using a single vs variable apriori profiles; it gives a homogeneous retrieval at the global scale and the retrieved variability is not distorted by that of the variable apriori. Hence, the true capability of the pair instrument/algorithm to capture the O3 variability is better infer when using a single apriori profile. That point should be clearly discussed in Sections 3.3 and 4.*

Here again we do not agree with the reviewer. A single a priori has long been used in SOFRID for the reasons mentioned by the reviewer and also because it is easier to build and easier to use. These are probably good reasons. Nevertheless, since that time some thorough O3 climatologies based on ozonesonde and satellite data such as Sofieva et al. (2014) have been made available and are used for other sensors (TES/OMI) retrievals as mentioned in the manuscript. They are the «most satisfactory» choice of a priori according to the textbook « inverse methods for atmospheric soundings » of Rodgers (2000). See for instance p166:

« The most satisfactory source of a priori information is from independent high spatial resolution measurements […] as may be obtaind from radiososnde measurements. Such data is often available as climatologies partitioned by, for example, latitude and date».

We fully acknowledge the stripes visible in the global distributions with V3.5. As mentioned in the manuscript they are due to the smoothing error with which they are in good quantitative agreement (< 5DU). If we look at tropospheric O3 in the SH mid-latitudes, we indeed see some marked stripes of about 2.5-5 DU which are of course errors. If we now look at the differences between SOFRID v1.6 and raw sondes we have an average bias of ~ 30% (Fig. 9e) while this difference drops to ~ 0% with V3.5 (fig. 11e). In the first case we have a smooth distribution with large but «invisible» biases and in the second case we have some visible effect of the a priori but low biases. We think that the retrievals are better in the second case. We have chosen to show the striped distributions to clearly document that issue for SOFRID data user.

There is also the improvement concerning the NH seasonal variability which is significant and very satisfactory. We think that these are not «wrong» improvements but just clear and documented ones. SOFRID is not IASI but a comprehensive system based on RTTOV and 1DVar algorithms with IASI radiances, ECMWF auxiliary data and a priori data as input. The validation exercice we have performed gives an evaluation of the whole system. For model comparisons there is no real issue because (i) the problem is already clearly acknowledged in the paper in case of comparisons with raw model data and (ii) as the IASI retrievals are the validating datasets (contrarily to here where they are the validated datasets) the modeled profiles should be smoothed by the retrieval AvKs which will take the issue of the variable a priori into account. To make things clearer, we added the following comment at the end of section 2.4 :

« Such stripes may appear as a problem for the use of SOFRID v3.5 data for model validation. They are a minor problem for two main reasons. First, as is demonstrated in next section, the use of a dynamical a priori largely improves the retrieved O3 profiles. Second, when model profiles are compared to SOFRID retrievals the impact of the a priori profile is taken into account by using Eq. 1 such as in Barret et al. (2016)»

*3/ Through Section 4, the authors insists on the fact that "the improvement of SOFRID accuracy . . .is the clearest advantage of using a dynamical apriori profiles". Given that several sources of improvement are taken into account: dependence on tropopause height, latitude and month, how can the authors be able to dissociate between their respective effects? Please, provide sensitivity tests or clarify that point? Comments 2/ and 3/ highlight the limitations in using variable apriori and evaluating the V3.5 product. The authors should better discuss those issues through the manuscript in order to get a better feeling for the real advantage of using variable apriori (in terms of both location,season and dynamical tropopause).*

In our introduction we define « a dynamical a priori profile for spaceborne O3 retrievals which takes the pixel location, time and tropopause height into account » and not only the tropopause. Therefore the statement mentioned by the reviewer about SOFRID improvements is correct and the important point is that the improvements are significant using such a « dynamical » a priori.

The validation has been performed in 30° latitude bands with monthly means. Therefore an important part of the answer is clearly in the paper. Indeed, the large improvement in the seasonal variability in the NH mid (and high) latitudes results from the monthly a priori. As these seasonal variabilities are latitude dependent it also highlights the importance of a latitude dependent a priori.

As the tropopause height is largely month- and latitude-dependent it is not possible and it would be artificial to fully « dissociate » the impact of the three parameters on the SOFRID improvements: a climatological a priori is implicitly tropopause dependent. Nevertheless, it is possible to assess the difference between a fully tropopause dependent a priori and a climatological a priori with an implicit tropopause dependence. This has been achieved with a sensitivity test with a single a priori (the one corresponding to the highest occurrence from Sofieva et al. (2014)) for each month and each 10° latitude bands therefore removing the intra-seasonal tropopause variability from the a priori choice.

The results are similar to those of the v3.5 highlighting that the improvement are little dependent on the intra-seasonal variability of the a priori profile. Nevertheless, v3.5 is better concerning the TOC variability in the 30-60°N band which is the most significant region in terms of sonde sampling and in the 60-90°S band. In the UTLS v3.5 is also better in terms of variability and correlation coefficients in most latitude bands.

Therefore, we have changed our manuscript in order to document the fact that the largest part of the improvement is due to the use of a climatological a priori dependent on month and latitude. This is of general interest for other scientists which are working on O3 retrievals and could use simpler climatologies. A new section (4.2 Impact of the intra-seasonal tropopoause dependence of the a priori profile on SOFRID improvements) including a figure with a Taylor diagramm presenting TOC and UTLS columns (Figure 6) has been added to illustrate this point.

We have also added a sentence in our conclusion:

«A sensitivity test demonstrated that these SOFRID improvements are dominated by the seasonal- and latitude- dependence of the a priori.»

*4/ Regarding the comparison with FORLI, the authors are very negative through the manuscript and the critics are most of the time out of context.*

We agree and we have improved the manuscript being more positive with FORLI. Nevertheless,  we would like to drow the reviewer attention to the fact that our initial statements were based on results published by the FORLI team.

*For instance:*
*- In the abstract: "(iii) in the N.H., no significant temporal drift is detected in SOFRID contrarily to FORLI (~8%)"*

This statement is based on Boynard et al. (2018) :
- in the text: «Based on the drift value with the 2σ standard deviation and the value (indicated on each plot), the derived **drifts** […] are **statistically significant** for the TROPO [...] columns (−8.6±3.4 % decade−1… )».
- in the conclusion : « A **significant negative drift** of −8.6 ± 3.4 % decade−1 is also found in the IASI-A to ozonesonde TROPO O3 column comparison for the Northern Hemisphere. »

Nevertheless, we have used « jump » instead of « drift » in the abstract.

« in the northern hemisphere, the 2010 **jump** detected in FORLI TOCs is not present in SOFRID ».

*- Introduction, L21: "They both document a problem (drift or jump) . . ."*

The full statement is «They both document a problem (drift or jump) in the O3 retrievals around year 2011 but this **does not hinder** the fact that TOC are decreasing according to Wespes et al. (2017). »
This is rather positive acknowledging the possibility to use the data for trends analysis as done in Wespes et al. (2017).

Following the reviewer recommendation about the use of « jump » rather than « drift », we have modified the statement as follows «…They both document a **jump** in the O3 retrievals in 2010 which does not hinder … »

*- Section 5, p.14, L.7-9: "the SOFRID NH tropospheric drifts discussed in section 4.3 are smaller and opposite in sign to the significant -8.6±3.4%/dec drift between FORLI and smoothed sonde data in the NH troposphere presented in B18."*
*That comparison of the "drift" calculated from SOFRID vs FORLI does not make sense. Indeed, the authors have to make a clear distinction between a "drift" that usually refers to an instrumental drift in validation studies, and a "jump" (or sudden discontinuity) as observed in the FORLI dataset, which induces an artificial drift, in order to avoid any confusion. It has already been clearly explained and discussed in Boynard et al. (2018) and in Wespes et al. (2018; 2019): the drift strongly decreases (< 1DU/dec on average) after the jump and it becomes even non-significant for most of the stations over the periods before or after the jump, separately. The discontinuity is strongly suspected to result from updates in level-2 temperature data from Eumetsat, which occur at the same date of the detected jump and which are used as inputs into FORLI. Hence, it is obvious that "No significant change occuring around 2010 is detectable for SOFRID v1.6 (Fig. 8(h)) and v3.5 (Fig. 10(h)) NH time series", given that SOFRID uses L2 from ECMWF, not from EUMETSAT. It should be clarified through the manuscript.*

As mentioned above, we based our comments and our mention of a **drift** on the recent papers concerning FORLI-O3 (Boynard et al., 2016, 2018 and Wespes et al. 2018). The two validation papers present the same 2010 «jump» even though it has not been clearly documented in Boynard et al. (2016). We understand the reviewer argument concerning the difference between a «jump» and a « drift » in FORLI O3 data. Nevertheless, it was not clear in the validation papers of the FORLI team (Boynard et al. , 2016 and 2018) and posterior publications. On the contrary:

- In the statement cited above from *Boynard et al. (2018)*, the words **statistically significant drift** are used.
- In *Wespes et al. (2018)* we read : « Note, however, that a **drift** in the NH middle-low troposphere (MLT) O3 over the whole IASI dataset is reported in Keppens et al. (2018) and Boynard et al. (2018) from comparison with O3 sondes. »

- in *Keppens et al. (2018)* : « Looking at latitude-resolved drift studies for the Ozone_cci IASI-A nadir ozone profiles (not shown), a **significant decadal negative drift** of the order of 25 % or higher can be observed in the Antarctic UTLS and the northern hemispheric troposphere. »

Concerning the cause of this jump, the reviewer mention «...i*s obvious that "No significant change [...] given that SOFRID uses L2 from ECMWF, not from EUMETSAT. It should be clarified through the manuscript. »*
We do not agree. The reason for the «jump» is not hypothesised as resulting from EUMETSAT L2 discontinuity in Boynard et al. (2018) and Wespes et al. (2018). More specifically :

- in the AMT Discussion of *Boynard et al. (2018)*, Reviewer #2 stated «Unfortunately the **significant drift** in the troposphere is barely explained and addressed». The authors replied « … a few more years are needed to confirm the observed **negative drifts** and evaluate it on the longer term… », statement which can be found in both the text and the conclusion of the final version of the paper.
- in *Boynard et al. (2018)*, the EUMETSAT L2 discontinuity is mentioned «It is worth mentioning that the EUMETSAT dataset is not homogenous, as it has been processed using different versions of the IASI Level 2 Product Processing Facility between 2008 (v4.2) and 2016 (v6.2)», but it is not clearly mentioned as an explanation for the TROPO-O3 drift/jump.
- in *Wespes et al. (2018) :* « This **drift** ($\sim$ 2.8 DU decade−1 in the NH) is shown in Boynard et al. (2018) to result from a discontinuity (called "jump"by the author) in September 2010 in the IASI O3 time series, for **reasons that are unclear** at present.»

Therefore, in these publications, no evidence (based on the EUMETSAT L2 products for instance) is given to explain the drift/jump.

It is in the two latest publications, *Keppens et al. (2018)* which is a general paper dealing with nadir ozone products and *Wespes et al. (2019)* dealing with Antarctic stratospheric O3, that the hypothesis of a causal link between EUMETSAT L2 and FORLI-O3 discontinuity is mentioned:
- *Keppens et al. (2018)*: «Part of the **overall negative tropospheric drift** of the FORLI v20151001 IASI retrievals **could, however, be due** to a change in the processing of the IASI L2 processor (e.g. temperature profile) at EUMETSAT that changed to version 5.0.6 in September 2010.»
- *Wespes et al. (2019)*: «The discontinuity is **suspected** to result from updates in level2 temperature data from Eumetsat that are used as inputs into FORLI (see Hurtmans et al., 2012). Hence, the apparent drift reported by Boynard et al. (2018) **likely** results from the jump rather than from a progressive "instrumental" drift.»

In these latest publication the words «**could be**», «**suspected**» and «**likely**» clearly mean that no formal evidence has been found to date.

Based on this review of FORLI litterature, we find that the possible explanation of the difference in calculated drifts for both algorithms with the EUMETSAT L2 products was not completely «**obvious**» for us at the time of writing our manuscript.

Nevertheless, we agree with Wespes et al. (2018) that a «jump» occuring in FORLI TROPO-O3 in September 2010 is responsible for most of the 8% «drift».
It is also noteworthy that the authors already discussed (i) the «jumpy» nature of the drift (ii) the role of this jump in the SOFRID FORLI difference concerning the NH tropospheric drift and (iii) even the potential link with EUMETSAT temperature at the end of the SOFRID-FORLI section a couple of lines after the statement cited by reviewer #2:
«These authors attribute their NH tropospheric significant drift to an abrupt change or jump detected in 2010 in FORLI [...] The difference could be linked to the use of EUMETSAT L2

products and of ECMWF analyses for FORLI and SOFRID retrievals respectively. As mentioned previously refering to B18, EUMETSAT L2 product are not homogeneous over the 2008-2016 period and a version change could result in the jump discussed in B18.»

In order to make things clearer we have modified this part taking the reviewer statements into account:

«Neverthelesss, the NH tropospheric drift from FORLI is attributed to an abrupt change or jump detected in 2010 (Boynard et al., 2018, Wespes et al. 2018). The drift strongly decreases after the jump and it becomes even non-significant for most of the stations over the periods before or after the jump, separately (Wespes et al. 2018). The discontinuity is suspected to result from updates in level-2 temperature data from EUMETSAT used as inputs into FORLI (Wespes et al., 2019). The absence of jump and the small drift in SOFRID v1.6 and v3.5 NH tropospheric data is therefore probably linked to the use of temperature profiles from ECMWF analyses instead of EUMETSAT L2 products.»

Finally, in the conclusion, we have modified the text to mention a jump rather than a drift in FORLI data:

« The difference with FORLI which is impacted by a significant TOC jump in 2010 (Boynard et al. 2018, Wespes et al. 2018) is likely linked to the use of different temperature profiles for the radiative transfer calculations (ECMWF analyses for SOFRID and EUMETSAT L2 for FORLI). »

*- Section 4.3, p.12, L.6-7: It has also to be clearly noted that Gaudel et al. (2018) study suffers from a lack of consistency between a series of parameters, such as the calculation of the tropopause, making the comparison not quantitative.*

In Gaudel et al. (2018) the tropopause is calculated according to the 2°K lapse rate from WMO for all of the satellite product. The different groups may have used different met-analyses (e.g. NCEP fo OMI-MLS, GOME and OMI, ECMWF for SOFRID, IASI-L2 for FORLI) but these resulted in rather little differences in tropopause height that cannot explain the significant differences in trends documented in this publication. In order to evidence the little impact of tropopause calculation in TOC trends and to compare our results with Boynard et al. (2018), we have computed the trends of the difference between sondes and SOFRID for both TOC and Surface-300 hPa columns. While the difference in column values is important (tropopause ~ 250-100 hPa versus 300 hPa), there are no significant changes (less than 0.2%) in the trends of the difference for the whole NH (see Fig 9 and 11 versus Fig. 16).

*- Section 5, p.12, L.32-33: First of all, on the contrary to what is stated in Section 3.4, three indicators (not only two) were calculated in Boynard et al. (2016, 2018), the fourth one (ratio of std) being rarely calculated in validation studies.*

We have corrected in the manuscript.

*That last one that makes possible to draw Taylor diagram is indeed interesting as it allows evaluating the representation of the retrieved variability. It could indeed be investigated for the validation of future FORLI products. Nevertheless, I am surprised that the authors did not perform their own analysis using the FORLI dataset that is publicly available on the french Ether/Aeris platform. It would have prevented possible inconsistencies between the SOFRID and the IASI datasets, the validation methodologies. . .*

As FORLI O3 data have been validated, we did not mean to re-validate them but to take advantage of the corresponding publications to check the consistency between datasets.

*For instance, in:*
*- Section 5, p.13, L9-10: One source of difference between FORLI and SOFRID could be the series of quality flags that have been applied on the datasets to select the best observations in terms of spectral fit and cloudy scenes. Are the flags comparable between the FORLI and the SOFRID datasets? Please comment.*

For SOFRID we filter the data with 3 quality flags. The first one described in the retrieval section (section 2) concerns clouds: we exclude cloud scenes based on the AVHRR cloud fraction cover. We now give the threshold which was missing:

« Pixels with AVHRR-derived fractional cloud cover larger than 25% are excluded»

Then we have three more data filters described in the validation section (Section 3): two based on the quality of retrieval (cost function Jcost > 0.0 for correct convergence and Jcost < 1.0 to elliminate worst fits), one concerning the information content (DFS > 2.0).

In order to show that the threshold values are not impacting substantially the SOFRID-FORLI comparisons we have performed sensitivity tests with different values. For Jcost we have used a trheshold value of 0.15 instead of 1.0 and the number of selected pixels decreased by 6%. Concerning the cloud filter, we have performed a test with 13% which is the value used in Boynard et al. (2018) instead of 25%. The number of pixels decreased by 5%. For the DFS we have made the comparisons with a trheshold of 1.75 instead of 2.00 (which is the trheshold used in Boynard et al. (2018)) and the number of pixels increased by 2%.

In each case, the general statistics changes are negligibles as can be seen in Figure 1 where the biases and RMSDs for v3.5 with the standard quality flags and with the modified trhesholds are presented.

[Figure]

Figure 1 : Biases and RMSDs of the differences between IASI retrievals and sonde data for the standard v3.5 (red), and v3.5 with modified quality flags: AVHRR cloud fraction cover (light blue), DFS (blue) and Jcost (green) (similar to Figure 14 in the paper).

The same is true for the correlation coefficients (r2) and slopes of the linear regressions (b) as shown in Fig. 2.

[Figure]

Figure 2. Slopes of the linear regression (positive values) and (-) r2 correlation coefficients (negative values) between IASI retrievals and sonde data (same as Fig. 13 in the paper).

The comparison of the two IASI-O3 retrievals presented in the paper is therefore robust and not highly dependent on the thresholds used to filter the data.
We have modified the text in the section corresponding to the SOFRID-FORLI comparisons accordingly adding the following text:

«Another limitation is that FORLI and SOFRID use their own quality flags to filter the data. In order to document the impact of the pixel selection on SOFRID validation we have performed the comparison with sonde data using modified quality flags. The cloud filtering trheshold is the clearest source of difference between the pixel selection of both algorithms. We have therefore lowered the upper limit of the AVHRR cloud fraction cover to 13% which is the trheshold used by Boynard et al. (2016, 2018) resulting in a loss of 5% of the treated pixels. The Jcost threshold has been decreased from 1.0 to 0.15 with a 6% decrease of the selected retrieved profiles. Finally the DFS lower value has been set to 1.75 increasing the number of selected retrievals by 2%. These

threshold modifications resulted in negligible changes of the general statistics (bias, RMSD, R) for the 3 atmospheric layers (troposphere, UTLS and stratosphere) and the different latitude bands that are presented in this section. These statistics, based on large numbers of data are therefore not hindered by pixel selection differences.»

*This is why taking data directly from literature for a quantitative comparison might be inappropriate and mislead the comparison. That issue/limitation in the comparison between SOFRID and FORLI should be clearly highlighted and discussed by the authors. I would strongly recommend the authors to better put the FORLI-SOFRID comparison into context with the reasons mentioned here above (i.e. jump in contrast with real drift, use of different quality flags, possible inconsistency between validation methodology. . .) through the manuscript.*

We agree with the reviewer that the SOFRID-FORLI comparison has important limitations because it is based on published results. Some limitations were already highlighted in the manuscript such as the fact that we could only compare results after smoothing of the sonde data. The jump issue has been largely discussed and amended throughout the manuscript as described above. The issue concerning data filtering is also now largely discussed in the manuscript with evidence given by sensitivity tests on the « quality flags ».  Nevertheless, it is important to realize that the quality flag issue is inherent to retrieval algorithm comparisons even with dedicated studies where the data are not taken from the literrature.

*- P.6, L.6-7: Why the behavior of TOC errors is similar to that of DFS while one can read above that the dominant source is the smoothing error? Please explain.*

The a priori variability is larger for the TOC in the tropics that at mid and high latitudes because of the higher tropopause height resulting in larger smoothing errors even with higher DFS. We have added the following explanation :
« This is due to the fact that the tropopause height is higher in the tropics resulting in a larger a priori variability. The impact of the increased variability exceeds the one of the increased information content resulting in a larger smoothing error »

*- P.10, L.2-3: Why does the smoothing of sonde profiles not improve the bias in UTLS while the DFS is < 1? Please explain.*

The application of Equ. 1 to the sonde profiles is supposed to correct biases linked to the a priori profiles independently from the DFS value. The fact that (i) the bias is present for v1.6 and v3.5 for which the a priori are different (ii) the application of Equ. 1 does not change significantly the bias, indicate that this particular bias (unlike the TOC biases) is not related to the a priori.
Therefore, this UTLS bias in IASI O3, already identified for the three retrieval algorithms (with and without smoothing) by Dufour et al. (2012) remains an issue.

*- Regarding the figures 12-14, one could think that the authors make their own analysis from the FORLI datasets, while the values are taken from previous validation papers. This should be clearly mentioned in the figure captions to avoid misunderstandings.*

We have added the ref to Boynard et al. (2018) in the captions.

Technical comments and typos:
*- P.2, l.22: The jump is detected in year 2010, not 2011.*
OK
*- P.2, L.30: tropospheric -> tropospheric*
OK

*- P.3, L.7: methodology -> methodology*

OK

*- P.4, L.33: "The use OF a . . ."*

OK

*- P.5, L.8: atmospheric -> atmospheric*

OK

*- P.6, L.1: Th -> The*

OK

*- P.7, L.20: one reference is missing here.*

We have added the ref Havemann (2020) for the convergence criteria of the NWP-SAF 1D-Var algorithm.

*- P.7, L.9: below -> above*

balloons with O3 sondes often explode below 40 kms.

*- P.7, L.21: elliminate -> eliminate*

OK

*- P.8, L.23: variance -> ratio of the variance (?)*

« ... is proportional to the variance of the experiment. Both RMSDs and standard deviations are normalised by the standard deviation of the reference... »

The second sentence implied that, after normalisation « the radial distance from the origin » is proportional to the ratio of the variance of the experiment to the variance of the reference.

We have changed the sentence to be clearer :

« We have normalised both RMSDs and standard deviations by the standard deviation of the reference to display the results from multiple experiments on a single diagram (see Taylor (2001) for details). »

*- Table 2: Units are missing*

We have added the units (%).

*- P.9, L.21: tropospehric -> tropospheric*

OK

*- P.9, L.27: UT -> UTLS*

OK

*- Fig.6 and 7: The legend is not clear. I guess RS means Raw Sondes and SmRS*

*means Smoothed Sondes. Hence, SmRS should be SmS (?). Please correct or clarify*

*in the caption.*

The caption has been clarified.

*- Error in the caption of Fig.9: "Same as Fig.9" -> "Same as Fig.8"*

OK

*- Fig.8: The color legend should be indicated in the top panels.*

The line colors are documented in the captions in order to avoid problems of legend superimposed on the lines.

*- P.12, L.6: Which version of SOFRID are you referring to?*

SOFRID v1.5 was used in Gaudel et al. (2018). We have added the version.

*- Fig.12 to 14 do not seem in correct order. Please consider this:*

*Fig.14 -> Fig.12, Fig.12 -> Fig.13, Fig.13 -> Fig.14*

We have reordered the citation to the figures in the text.

*- P.13, L.1: delete "(b)" in the sentence. I don't see that in Fig.13.*

We have modified the caption adding the ref to « b ».

---

## Author Response (AR2)

Dear Editor,

We have performed the 2 minor changes required by reviewer #1.
She suggested to change the title with "a comprehensive dynamical a priori…" or something equivalent rather than "a tropopause-based a priori…".
Our a priori is based on the paper by Sofieva et al. « A novel tropopause-related climatology of ozone profiles » and is presenting improvements and validation based on the use of this a priori. We demonstrate that most of the improvements come from the climatological nature of the a priori, and that a smaller part comes from the tropopause dependence.
We therefore changed the title to «A climatological tropopause-related a priori for IASI-SOFRID Ozone retrievals: improvements and validation » because that is precisely what the paper is dealing with.
In the same way, we have modified the text in the conclusuion (p16 l1) with « The use of a climatological tropopause-related a priori... » instead of just « tropopause-based » as previously.

Finally, we have corrected the typos indicated by the reviewer.

Best regards,
Brice Barret